# High resolution time series reveals cohesive but short-lived communities in coastal plankton

Antonio M. Martin-Platero[1,6], Brian Cleary[2,3], Kathryn Kauffman [ID] [1], Sarah P. Preheim[1,7], Dennis J. McGillicuddy, Jr[4], Eric J. Alm[1,2,5] & Martin F. Polz[1]

Because microbial plankton in the ocean comprise diverse bacteria, algae, and protists that are subject to environmental forcing on multiple spatial and temporal scales, a fundamental open question is to what extent these organisms form ecologically cohesive communities. Here we show that although all taxa undergo large, near daily fluctuations in abundance, microbial plankton are organized into clearly defined communities whose turnover is rapid and sharp. We analyze a time series of 93 consecutive days of coastal plankton using a technique that allows inference of communities as modular units of interacting taxa by determining positive and negative correlations at different temporal frequencies. This approach shows both coordinated population expansions that demarcate community boundaries and high frequency of positive and negative associations among populations within communities. Our analysis thus highlights that the environmental variability of the coastal ocean is mirrored in sharp transitions of defined but ephemeral communities of organisms.

[1] Department of Civil and Environmental Engineering, Massachusetts Institute of Technology, Cambridge, MA 02139, USA. [2] Broad Institute, Cambridge, MA 02139, USA. [3] Computational and Systems Biology Program, Massachusetts Institute of Technology, Cambridge, MA 02139, USA. [4] Department of Applied Ocean Physics and Engineering, Woods Hole Oceanographic Institution, Woods Hole, MA 02543, USA. [5] Department of Biological Engineering, Massachusetts Institute of Technology, Cambridge, MA 02139, USA. [6] Present address: Department of Microbiology, University of Granada, Granada 18071, Spain. [7] Present address: Department of Geography and Environmental Engineering, Johns Hopkins University, Baltimore, MD 21218, USA. Antonio M. Martin-Platero and Brian Cleary contributed equally to this work. Correspondence and requests for materials should be addressed to E.J.A. (email: ejalm@mit.edu) or to M.F.P. (email: mpolz@mit.edu)

Because microbes in the surface waters of the ocean have key roles in global carbon and nutrient cycling, they have been under intense scrutiny, revealing an increasingly complete picture of their global taxonomic composition and functional repertoires (e.g., refs.[1–3]). What remains poorly understood, however, are spatio-temporal dynamics of the vast majority of taxa, which include diverse representatives from all domains of life, pursuing different ecological strategies. These include bacteria forming complex biofilms on different types of organic particles or existing as free-living, single cells, and photosynthetic, filter-feeding, and predatory eukaryotic plankton. For these organisms, ocean water represents a nutrient poor but highly heterogeneous ecological landscape on different spatial and temporal scales with physical and chemical gradients ranging from micrometers to kilometers[4–6]. For example, direct competitive or cooperative interactions may lead to rapid micro-scale successions on suspended organic particles[7], whereas large-scale algal blooms may trigger growth of bacteria that degrade specific algal exudates[8]. Owing to the apparent difference in scales of such interactions, it remains unknown whether communities with clearly defined boundaries in time and space can be identified across the entire plankton, or whether taxonomic turnover is more gradual and continuous, affecting limited groups of organisms at a time.

Analyses of time series of microbial plankton have to date been equivocal towards the question of organization into cohesive communities. Although longer but sparsely sampled time series have provided evidence for seasonal recurrence and successions of some microbial taxa (e.g., refs.[9,10]), such patterns have so far proven weak across the entire plankton and have provided no indication of sharp transitions[11–13]. In fact, more densely sampled but short time series have revealed brief and intense fluctuations in relative abundance of operational taxonomic units (OTUs)[14–16], which are the most highly resolved taxonomic units in ribosomal RNA-based diversity studies. However, because current time series have either been longer but sparsely sampled or densely sampled but short, their resolution is limited over time scales relevant for both detecting microbial growth and transitions in ecological conditions so that associated community change had been difficult to constrain. We therefore reasoned that to assess the extent of organization into communities over time, an extended high-resolution time series of bacterial and eukaryotic plankton was necessary that could capture both rapid changes due to organismic interactions as well as longer range dynamics due to transitions in overall ecological conditions.

Here we sample and analyze coastal bacterial and eukaryal plankton over 93 consecutive days and show that although individual taxa fluctuate on near daily time scales, the amplitude of these fluctuations is characteristically greater during limited time periods, indicating preferential growth. To determine whether such temporally limited expansions are coordinated across multiple taxa as expected from cohesively behaving communities, we use wavelet-based analysis to establish correlations over different time spans. This analysis shows that highly cohesive communities can be identified that are differentiated by environmental and organismal features and that turnover rapidly on the order of a few days.

## Results

**Contrasting plankton dynamics at different taxonomic resolution.** To determine the dynamics of bacterial and eukaryotic plankton over varying temporal scales, we collected daily water samples from a coastal ocean site (Nahant, MA, US East Coast) over three consecutive months spanning a summer to fall seasonal transition (23 July to 23 October, 2010; see Methods).

Because we use fixed-point (Eulerian) sampling, as has been standard in ocean time-series studies, our data integrate temporal and spatial components of variability associated with the continuous movement of water masses via tidal cycles and ocean currents. To assess organismal diversity, we carried out Illumina-based tag-sequencing of bacterial 16S and eukaryotic 18S rRNA genes followed by identification of OTUs (Methods). To maximize ecological resolution of OTUs, we used distribution-based clustering (dbc), which does not assume a fixed sequence similarity cut-off to define OTUs but instead identifies the sequence similarity at which clusters display cohesive behavior across samples[17]. This approach usually yields clusters comprising very closely related sequences and resulted in a total of 49,637 bacterial and eukaryotic OTUs, of which 9660 reoccurred on more than 10 days. These recurrent OTUs were used to test the extent to which plankton are naturally organized into clusters of taxa with correlated dynamics as expected for communities of interacting organisms.

Initial analysis of the time series indicated relatively stable dynamics across higher taxonomic ranks (phylum to family), but very rapid, near daily fluctuations among both bacterial and eukaryotic OTUs (Fig. 1). Although a similar pattern was noted in a shorter daily time series of coastal plankton on the US West Coast[16], our longer time series revealed another pervasive pattern: although many individual OTUs tended to persist at low relative abundance across the entire time series, they frequently showed limited periods of expansion, during which they rose to higher relative abundance (on average, across the period of the expansion), before returning to a low level. Such patterns have previously been described as seed bank dynamics[18,19] and Fig. 2 shows a typical example of such variation of a eukaryotic primary-producer (diatom) and bacterial heterotrophic (Flavobacteria) OTU, which both fluctuated at low levels during most of the time series but displayed higher amplitude of fluctuation during a limited period. These dynamics are consistent with ecological forcing acting at multiple temporal scales, wherein baseline carrying capacity for an OTU gradually increases at larger temporal and spatial scales but biological interactions might induce shorter term fluctuations due to cooperation, succession, competition, and predation[20–23]. We therefore hypothesized that communities of interacting organisms should show coordinated expansions of OTUs over longer periods along with more rapid fluctuations driven by direct positive or negative biotic interactions, together with environmental forcing.

**Identifying microbial communities amidst high OTU variation.** To allow for detection of such coordinated longer expansions coupled to shorter term biotic and environmental interactions, we developed a novel clustering method based on wavelet analysis (WaveClust). As in previous applications to ecological time series[22,24], wavelet analysis allowed us to decompose each OTU time series into lower and higher frequency components, while maintaining both temporal resolution and phase information. This enables identification of temporally coordinated periods of low-frequency expansion in addition to dynamics either due to positive or negative interactions at higher frequency where interactions here denote correlated dynamics. Given the information in each decomposed time series, we were specifically interested in defining pairwise interactions between OTUs based on correlation or anti-correlation at multiple distinct frequencies, and in using a network of such interactions to identify emergent communities through application of a Markov clustering algorithm[25] (Methods).

WaveClust advances beyond earlier applications of wavelet decomposition in macroecology[22,24] by considering a large

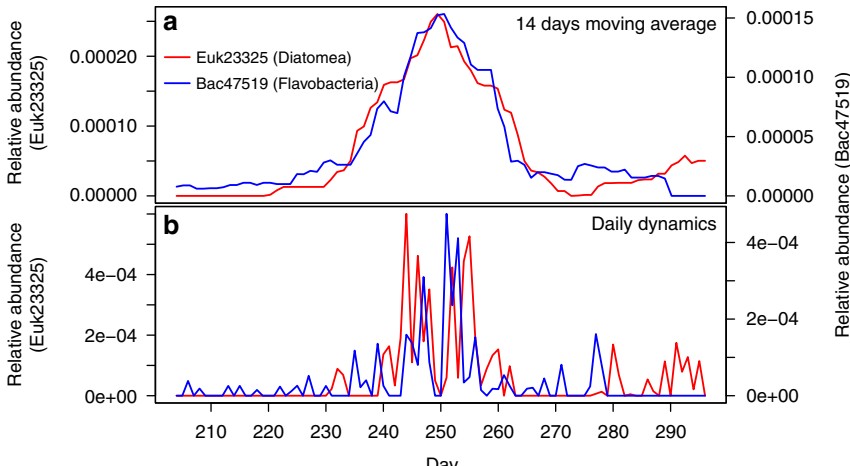

**Fig. 1** Contrasting dynamics of plankton at different levels of taxonomic resolution. Although relative abundances of bacteria (**a**) and Eukarya (**b**) appear relatively stable when evaluated at the phylum to family level, they vary extensively and rapidly at fine-scale taxonomic resolution (OTU-level) (**c**), also evidenced by different rates of decline in taxonomic similarity (1-Jensen–Shannon distance) for phylum- and OTU-level with increasing time lags (**d**). Legends for phylum and family-level taxa are ordered from most to least abundant. Error bars in **d** denote standard error of three independent samples per day

**Fig. 2** Example of limited period of higher average abundance of OTUs. For the diatom OTU Euk23325 and the Flavobacteria OTU Bac47519, **a** the 14 day moving average and **b** daily dynamics are shown. On short time scale (days) both OTUs are negatively correlated, whereas over longer time scales (weeks) they are positively correlated

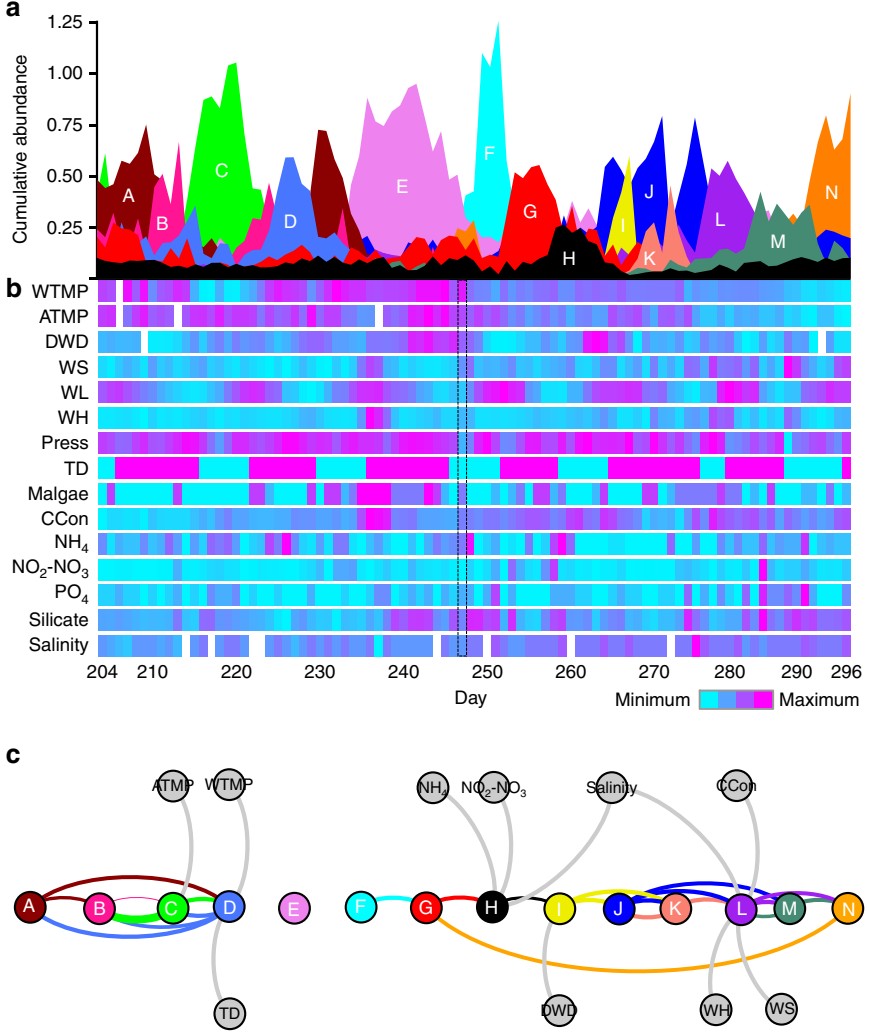

**Fig. 3** Dynamics of predicted communities and metadata across the time series. **a** Communities predicted by WaveClust analysis as modular units (clusters) of interacting OTUs based on wavelet decomposition to determine correlations at different temporal frequencies followed by clustering (Methods). Shown are the results from positive correlation at both low and high frequency. **b** Heat map displaying change in physical, biological, and chemical environmental parameters over the time series. Color scale for each environmental parameter varies between maximum and minimum values, which are for each parameter: WWTMP: 20–10 °C; ATMP: 30–10 °C; DWD: 14–4S, WS: 14–4 m/s; WL: 4.5–1.5 m; WH: 3–0.5 m; Press: 1020–995 hPa; TD: 1–0 incoming/outgoing; Malgae: 4–0 relative concentration; CCon: 10–4 µg/L; NH$_4$: 1.4–0 µM; NO$_2$-NO$_3$: 20–0 µM; PO$_4$: 0.7–0 µM; Silicate: 10–2 µM; Salinity: 36–33 psu. For more details see Supplementary Data 7, where values for each day are listed. Day 247, which marks Hurricane Earl passage, is framed in the heat map. **c** Granger causalities linking predicted communities to each other and to environmental parameters. Nodes represent communities (colored according to **a**) and environmental parameters (gray). Legends for **b** and **c**: WTMP, water temperature; ATMP, air temperature; DWD, dominant wave period; WS, wind speed; WL, water level; WH, wave height; Press, pressure; TD, tidal direction; Malgae, macroalgae; CCon, chlorophyll concentration; NH$_4$, ammonium; NO$_2$–NO$_3$, nitrite and nitrates; PO$_4$, phosphate

number of high-resolution interactions (i.e., at the OTU-level as compared with the guild level) in a highly complex environment, and by generalizing frequency-interaction analysis to capture distinct (and possibly different) relationships at high frequency and low frequency. The emergent global structure of the interaction network is then used to identify clusters of interacting OTUs as communities. Each community is defined using a graph clustering algorithm that finds densely interconnected OTUs, using both high-frequency and low-frequency correlations to define edge weights (Methods).

Simulations confirmed that WaveClust was able to correctly identify OTU time series that were coupled at both high frequency and low frequency, whereas standard Pearson correlation analysis was not sufficient to capture these coupled dynamics (Supplementary Fig. 1, Methods). Simulated data were also used to assess how peak bloom size, number of blooms, and the period

of blooms may affect results (Supplementary Fig. 2A). We found that positive associations (between a pair of OTUs that are correlated at both low and high frequency; Supplementary Fig. 2B) are more difficult to detect with our method than negative associations (between OTUs that are anti-correlated at high frequency; Supplementary Fig. 2C), that our method is less sensitive with increasing bloom abundance, and that results were largely robust to the period of the bloom. This was true whether we allowed for only one bloom per time series (as in representative examples in Supplementary Fig. 2A), or multiple blooms.

**Defined communities with rapid and sharp turnover.** Application of WaveClust to our daily time series of bacterial and eukaryotic plankton revealed that associations between taxa over

time were highly organized and comprised of distinct clusters of OTUs as expected for cohesively behaving communities of interacting organisms. These communities turned over very rapidly, reaching peak abundances on the order of only a few days. However, communities typically persisted at lower abundances across the time series with occasional recurrence of peaks (Fig. 3a). This demonstrates that nearly all OTUs across bacteria and Eukarya can be rare over extended periods of time and that dominance may change rapidly over small temporal and/or spatial scales. These observations of rapid community-level turnover run counter to the intuition that there should be consistent seasonal trends in community composition and may explain why compositional similarity in longer but more sparsely sampled time series decays quickly even when the same season in consecutive years is compared[11,12].

We found highly consistent community-level organization when considering either positive (Fig. 3a) or negative (Supplementary Figs. 3A, 4, and 5) high-frequency interactions, suggesting both types of interactions are pervasive within communities. Consistent organization is supported by (i) high overlap in OTUs among positive and negative high-frequency clusters (Supplementary Fig. 5) and (ii) a mutual information score (a quantitative measure of cluster agreement) of 52% between the two sets of clusters (the mutual information with randomly generated clusters is 0%). The most consistent difference was that several larger clusters in the negative-interaction analysis broke into pairs of clusters in the positive interactions (Supplementary Figs. 4 and 5; Supplementary Table 1). This indicates that there were characteristic high frequency negative interactions (possibly due to competition or predation) between pairs of OTUs occurring over longer time periods but that high-frequency positive interactions (possibly due to cooperation) were temporally more limited. One explanation of the latter would be that shorter lived blooms of organisms exploiting similar resources may lead to metabolic interactions (Fig. 3a; Supplementary Fig. 3A). Although it has been more common to use positive correlations to estimate community-level interactions, the high agreement with negative high-frequency correlation-based community structure estimates suggest that both follow similar organizing principles. In other words, positive correlation over longer periods indicates that changes in overall ecological regimes select for consistent sets of organisms, which then engage in the predicted positive and negative interactions over shorter time scales.

**Testing robustness of community predictions**. To assess the statistical significance of the observed clustering of taxa into communities, we devised a permutation-based test. We permuted each time series independently and calculated wavelet similarity scores, noting for each pair of OTUs if the scores were greater than those observed in the original data. We repeated these permutations 50,000 times, and used the results to calculate empirical $p$-values. This analysis revealed that the real data contain significant structure not present in shuffled time series: the original data have a clear enrichment for significant $p$-values (Supplementary Fig. 6A); clusters that only use connections passing a false discovery rate (FDR) of 10% are in good agreement with our original results (55% mutual information); and, if we repeat the FDR and clustering analysis using shuffled time series in place of the original, we observe very little clustering (just a single cluster containing only 122/9660 OTUs). We also note that using the conservative FDR test, only 5% of the scores that passed the original thresholds also pass the FDR test. Thus, overall cluster structure remains robust, even if less stringent connections between nodes are allowed.

Moreover, the clustering of taxa into communities was robust to the potential effect of autocorrelation at short time scales, and to how OTUs were defined (i.e., by distribution-based clustering vs. uniform similarity cut-off). Although overall population similarity at short time scales is high (~50–60% with a 1-day lag), especially relative to longer (e.g., 1 month) time scales (Fig. 1d), the autocorrelation of each time series with a 1-day lag is modest (median of 19%; Supplementary Fig. 7A). Using first difference data to remove the autocorrelation, we found that high-frequency similarity scores were relatively unchanged (86% correlated with scores from the original data; Supplementary Fig. 7B), and that the resulting clusters are highly similar to the original clusters (81% mutual information for positive interactions, 82% mutual information for negative interactions). Using a different way to define OTUs (i.e., simply grouped at 100% sequence identity), we found significant $p$-values at a frequency similar to the original data (Supplementary Fig. 6B), and also found that clustering of non-distributional OTUs results in a similar number of clusters, sharp transitions, and similar patterns of abundance (Supplementary Fig. 8). We note that it will be important to keep in mind the potential effects of autocorrelation, especially when there is a large difference in autocorrelation between long and short time scales. Although our results were robust to these potential effects, this should not be taken for granted in new time series, and analysis should be performed on differenced data.

**Fine-scale taxonomic differentiation of communities**. A key attribute of the time series, consistent with the detection of changing ecological conditions, is the fine-scale taxonomic differentiation of taxa among communities. First, the time series followed a general pattern of alternating predominance of diatoms and dinoflagellates (Supplementary Figs. 9, 10, and 11), which both have distinct nutrient requirements and both occur under differing environmental conditions[26]. However, further niche partitioning within these broad groups of phytoplankton is suggested by closely related OTUs peaking in different communities (Supplementary Fig. 12). Second, peaks of primary-producer OTUs frequently correlated to rapid expansions of specific bacterial OTUs (e.g., Fig. 2), a pattern that has previously been shown to be due to successions of bacteria capable of degrading different algal exudates[8,10]. Our data suggest that these associations might be specific at the OTU-level so that rapid turnover at the primary-producer level may lead to taxonomically highly resolved bacterial successions via trophic interactions. Overall, these patterns evoke Hutchinson's paradox of the plankton hypothesis, which postulates that phylogenetic diversity of phytoplankton exceeds the number of necessary resources because physical structuring and biological interactions creates additional niche dimensions[27]. Consistent with this hypothesis, the temporally highly structured diversity into communities of interacting organisms suggests that the realized niche space is relatively narrow, and may result in low ecological redundancy even among closely related primary-producer OTUs and the bacteria they interact with.

The above illustrates a subtle but important point for interpretation of community assembly. The most closely related OTUs are more frequently associated with different communities. When comparing the distribution of OTUs defined by varying genetic distance from 0 to 0.06 across all bacterial and eukaryal taxa, the majority of pairs fall into different communities (Fig. 4). Such differential dynamics of close relatives is generally interpreted as negative interactions (e.g., competition) being important in community assembly[28]. It may also be an indication of frequent speciation leading to niche partitioning. This

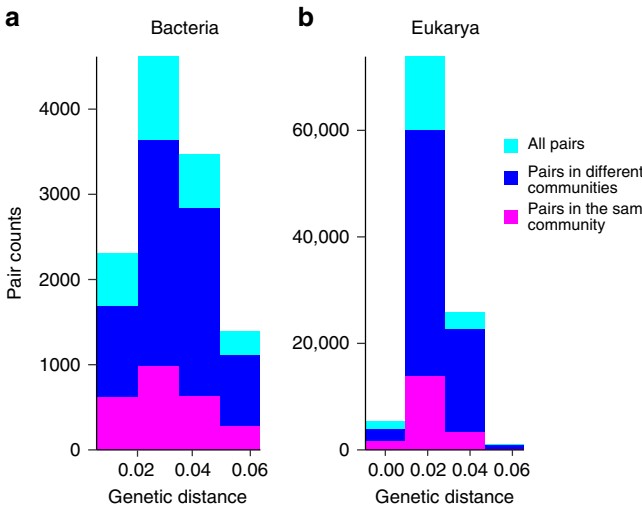

**Fig. 4** The most closely related OTUs tend to peak in different communities suggesting ecological differentiation. Shown are overlapped histograms of pairs of representative dbc-OTUs falling within the same (blue bars) or different (red bars) communities. Green bars represent the sum of blue and red bars. When OTUs are defined with high sequence similarity (>97%), they preferentially occur in different communities, for both bacteria (**a**) and Eukarya (**b**). Observed distributions were significantly different from a random community membership assignment ($\chi^2$ goodness of fit tests: $\chi^2$ (1) = 3042, *p*-value < 2.2e−16, for bacteria; and $\chi^2$(1) = 14,735, *p*-value < 2.2e−16, for Eukarya)

interpretation is consistent with our previous findings of differential microhabitat association and ecological specialization of closely related *Vibrio* populations that are all identical in 16S rRNA genes[29,30]. In fact, such differential associations may be a direct outcome of the speciation process as the formation of distinct lineages in a highly interconnected environment like the ocean requires the evolution of ecological tradeoffs, which induce spatial and/or temporal separation of gene pools and hence allow differential ecological specialization to evolve and adaptive traits to spread in a population-specific manner[31]. The strong pattern of differential distribution of the most highly resolved taxonomic units might thus indicate that speciation is occurring frequently and cautions that using too broadly defined taxonomic units might miss important ecological dynamics.

**Environmental changes associated with community transitions**. Assessment of WaveClust clusters in light of observed environmental conditions provides further strong support for different physical, chemical, and/or biotic conditions structuring the identified communities. First, inspection of satellite data of regional chlorophyll *a* (chl *a*) concentration and sea surface temperature (SST) suggest that some of the fluctuations in communities coincide with transient mesoscale features such as upwelling water masses and eddies (Supplementary Fig. 13). Second, network analysis of the Granger causalities between clusters and metadata, and also among clusters, revealed two main modules, which represent distinct chronological "meta-regimes" in the times series and were differentiated by shifts in nutrients, macroalgal abundance, and physical parameters (Fig. 3c; Supplementary Fig. 3C). Notably, both meta-regimes were characterized by alternating fluctuations in dominant communities that were disrupted by two major disturbances.

The first period (represented by clusters A to D in Fig. 3c, and A to D and O in Supplementary Fig. 3C) captures a seasonal

warm period disrupted by the abrupt movement of a distinct water mass into the coastal sampling site. Eleven days into the time series the prevailing communities (clusters A and B) were suddenly replaced by another highly abundant community (cluster C) accompanied by a drop in water temperature over 2 days from 18.3 to 10.9 °C, likely due to an upwelling event (Fig. 3a, b; Supplementary Figs. 3A, 10 and 13). Notably, this appeared as a shift from a dinoflagellate to diatom-dominated community (Supplementary Figs. 9 and 10) consistent with previous observations that the latter are associated with higher nutrients conditions typical of upwelling events[32,33]. This interpretation is reflected in chl *a* and SST data obtained by satellite. Although these data are patchy due to variable cloud cover, they suggest an initial low chl *a*, high SST period that transitions to a high chl *a*, low SST feature along the coast by day 221 (Supplementary Fig. 13). With the subsidence of the upwelling mass the site returned to initial temperatures and the two original communities rebound briefly albeit separated by another short-lived community (cluster D).

The second meta-regime (starting around day 235) represents an initial warm period marked by the passage of a hurricane system and followed by gradual, seasonal cooling of the water (Fig. 3c; Supplementary Figs. 3C and 14). Preceding the hurricane is a period with high macroalgae abundance and increased wave height that coincides with the emergence of cluster E, which notably contains a strong *Vibrionaceae* bloom (Supplementary Fig. 15) that may be driven by their positive correlation with higher water temperature and their ability to degrade brown algal polysaccharides[30,34]. This warm and high chl *a* period (Supplementary Fig. 13) came to an abrupt end with the passage of Hurricane Earl on ordinal day 247, with near-coastal SST dropping significantly due to storm-induced mixing and/or upwelling. Community F subsequently spiked, likely due to the passive resuspension of benthic material and sediment as this community contained a high proportion of taxa, such as nematodes, not typically found in the water column. Indeed, high incoming tides and below average dominant wave period indicate elevated currents over a large area of shallow sediments (Fig. 3c).

Finally, the increase and eventual dominance of cluster G, which is enriched in diatoms and dinoflagellates, as well as Flavobacteria (Supplementary Fig. 9), coincided with a sudden drop from ~20 to 16 °C in water temperature but regional high chl *a* following Hurricane Earl (Supplementary Fig. 13). The abrupt end of this algal bloom around day 261 is marked by a transition to the short-lived cluster H during a lower chl *a* period, which is subsequently replaced by a short but intense algal bloom on days 264 through 266 coincident with the expansion and collapse of cluster I. Subsequently, cluster L develops during a low chl *a* feature that spreads along the coast and disappears around day 286 after which chl *a* increases indicating the development of a new algal bloom at the same time as cluster N appears (Supplementary Fig. 13). The continued gradual decrease in water temperature during this latter period in the time series is consistent with seasonal transition to fall conditions (Supplementary Figs. 13 and 14). Although such gradual change might be expected to correspond to a gradual change of taxa, there was instead very rapid turnover of several communities with a notable repeated recurrence of one community (cluster J). Hence, although the temperature was similar to the cold-water intrusion in the beginning, different communities dominated this seasonal cooling event demonstrating that although temperature is an important master variable, combinations of other factors, including physical variability and other environmental factors, trigger changes in entire communities rather than individual taxa.

## Discussion

Our time series combined with novel analysis techniques (WaveClust) reveals surprisingly clear organization of coastal plankton into communities defined as modules of correlated taxa. These communities are likely assembled by coordinated expansions of OTUs in response to favorable growth conditions modulated by both positive and negative interactions among OTUs and the environment, leading to rapid fluctuations of taxa. In fact, our analysis shows surprisingly high overlap in communities defined by positive and negative interactions at high frequency (Fig. 3; Supplementary Figs. 3, 4 and 5), suggesting that both affect OTU dynamics within the same communities.

These organismal interactions may strongly contribute to high specificity in community membership detected here with even the most closely related OTUs primarily occurring in different communities, suggesting that they are ecologically differentiated. These observations pose the intriguing question to what extent communities reassemble across longer time periods as further evidence of their cohesiveness and whether they assemble reproducibly enough for their OTU composition to serve as an accurate indicator of environmental conditions. Answering this question will require expanding sampling to multiple years and other ocean regions. These may show slower dynamics than the highly productive and dynamic coastal ocean; however, our results stress that more time series are required that combine both high-frequency sampling over shorter periods with longer, low-frequency sampling to answer the question of community turnover in the ocean.

Because our sampling regime captured elements of both spatial and temporal heterogeneity, the observed rapid turnover of communities suggests they are highly patchy at either short temporal and/or small spatial scales. Indeed, such intermittency and patchiness is characteristic of the global coastal ocean[35]. However, the spatial scale of the types of plankton communities described herein remains unknown, and spatial surveys are needed to characterize this patchiness. Although we cannot at this point clearly disentangle spatial from temporal variation, inspection of satellite data support rapid change in local oceanographic features such as upwelling water masses and eddies that may provide short-lived but stable conditions for different communities to form (Supplementary Fig. 13). The transitions in communities in the beginning of the time series were most likely due to changing wind patterns inducing upwelling events (Fig. 3). However, later in the time series there were more gradual, seasonal changes of the water that may have led to community turnover within a more similar body of water. Irrespective of the reason, such rapid turnover in microbial communities is highly relevant for larger sessile and mobile organisms as they will lead to frequent encounter of different sets of microbes that may include different food but also pathogens and other harmful organisms. Therefore, the observed rapid change in OTUs and communities on daily time scales also bears relevance for monitoring of the coastal ocean for recreational and commercial human use. Such monitoring may require high frequency of biological sampling in order to critically evaluate rapidly changing potential exposures. However, paraphrasing Mark Twain who once famously said that "if you don't like the weather in New England, just wait for a few minutes", the good news may be that if you don't like the microbial community, just wait for a few days.

## Methods

**Environmental sampling and metadata**. Samples were collected at Canoe Cove, Nahant, MA, USA (Lat: 42° 25′ 10.6″ N, Lon: 70° 54′ 24.2″ W) between 09:30 am and 11:30 am (median 10:40 am) every day between July 23, 2010 (ordinal day 204) and October 23, 2010 (ordinal day 296). Nahant is a 2.7 km² rocky peninsula connected to the mainland by a ~3 km long causeway. This site was chosen because there is no large freshwater input on the peninsula so that the daily water samples represent coastal water in the Massachusetts Bay as it moves driven by currents and tidal cycles. The absolute location of sampling at Canoe Cove varied each day dependent on tidal height. Samples were always taken in triplicates for each day. Between Day 204 and 258, the three samples were taken at the same location, from Day 259 forward replicates for each day were collected separately at three spatially separated stations to test for variation on small spatial scales (~20 m). Independent of the sampling scheme, variation between consecutive days was always much larger than within days (Kruskal–Wallis $\chi^2 = 228.63$, df = 87, $p$-value = 1.145e−14 for the entire series, and Kruskal–Wallis $\chi^2 = 76.722$, df = 33, $p$-value = 2.448e−05 for the spatially separate stations). Associated metadata values, as described below, are provided in Supplementary Data 7.

Air and water temperatures were measured on site each day. Salinity was measured for water samples collected each day on stored water samples using a Reichert digital refractometer calibrated to a standard salt solution.

Water samples for DNA extraction were collected each day as triplicate 4L samples in autoclaved screw-cap polypropylene bottles. Upon return to the lab the water from each of the 4L samples was filtered by peristaltic pump from the collection vessel through 0.2 μm Sterivex filters (Millipore, SVGP01050) and the filters stored at −20 °C; time between sample collection in the field and storage of filters was ~3.5h.

Water samples for analysis of ammonium, phosphate, combined nitrate and nitrite, and silicate were collected in 250 mL polypropylene bottles, filtered through ashed GF/F filters (Whatman, 1825−047), collected into 20 mL scintillation vials and stored frozen daily. All materials used to process nutrient samples were acid washed prior to use. Nutrients were analyzed by the Nutrient Facility at Woods Hole Oceanographic Institution following completion of the field sampling study. Correction factors were applied for all nutrients in cases where volume loss occurred during freezing, correction factors for silicate were assumed to be linear with change in salinity as previously shown for phosphate and nitrate[36].

The abundance of macroalgae in the water during each daily sampling at Nahant was ranked on the basis of daily field notes and photographs following the completion of the study.

Water level and tidal direction data sets were based on water level data available for nearby NOAA Boston Harbor station 8443970. Verified 6-min water level data for each month of the time series were downloaded from: http://opendap.co-ops.nos.noaa.gov/axis/webservices/waterlevelverifiedsixmin/index.jsp. A new vector for tidal direction was generated assigning each time point as either "Incoming" (1) or "Outgoing" (0), depending on whether the water level was higher or lower than the previous 6-minute time point, respectively. The time point for each day differed and was dependent on arrival and departure time from the sampling site each day.

Several data sets from NOAA Station 44013, located 16 nautical miles east of Boston, MA, were used to assess relation of Nahant observations to larger scale regional conditions, which were: chlorophyll concentration, atmospheric pressure, dominant wave period, wave height, and wind speed. These data types were contained within historical records for "Ocean Data" (chlorophyll concentration) and "Standard Meteorological Data" (all others), which were downloaded from http://www.ndbc.noaa.gov/station_history.php?station=44013. Data were processed as follows: Times were converted from UTC to EST; all no-data values (e.g., "99", "999" or "9999") were removed; 24 h averages were calculated for each time point by averaging values for parameter at a given time point and all time points in the previous 24 h; time points for days with greater than 10 missing hourly values in the preceding 24 h window were discarded. The time point used for each day was 10:50 am EST for all data types except chlorophyll concentration, for which 11:00 am EST was used.

Satellite sea surface temperature and chlorophyll $a$ fields were obtained from Level 2 MODIS-A browser at https://oceancolor.gsfc.nasa.gov/cgi/browse.pl?sen=am. Scenes were relatively cloud free over the domain of interest and compiled for the period July–October 2010.

Information regarding the passage of Hurricane Earl, which crossed the study site latitude between 02:00 am and 08:00 am on 4th September (ordinal day 247), was obtained from the National Hurricane Center[37].

**Nucleic acid extraction**. To isolate genomic DNA from the 279 water samples (93 days in triplicates) we used a bead beating approach. Filters were sterilely cut and placed into 2 mL screw-cap tubes, followed by the addition of 0.25 g of sterile 0.1 mm zirconium beads (BioSpec Products Inc., Bartlesville, OK). Cells were lysed by adding 750 μL of Cell Lysis Solution (Qiagen, USA) and shaken in a Mini Beadbeater-1 (BioSpec Products, Inc., Bartlesville, OK) at 5000 rpm for 60 s, followed by incubation at 80 °C for 5 min. Once samples were cooled down to room temperature, RNA was digested by adding 4 μL RNAse A (4 mg/mL), mixed by inverting the tubes, and incubated at 37 °C for 30 min. DNA was purified by adding 250 μL of Protein Precipitation Solution (Qiagen, USA), mixed by vortexing for 20 s, incubated on ice for 5 min and centrifuged at 13,000 rpm for 5 min. Supernatant was transferred to a new tube and centrifuged again to ensure removal of all precipitates. Subsequently, 750 μL of isopropanol were added to 750 μL of supernatant to precipitate DNA, and incubated at −20 °C overnight. DNA was recovered by centrifugation at 13,000 rpm for 5 min, and washed with 700 μL of 70% ethanol. Finally, the DNA was dried and resuspended in 100 μL of DNA hydration solution (Qiagen, USA).

**Ribosomal RNA gene amplification and sequencing**. To generate 16S (bacteria) and 18S (Eukarya) ribosomal RNA gene libraries for tag-sequencing, we constructed paired-end Illumina libraries using a two-step PCR approach[17]. In the first step, we used primer pairs targeting the V4 variable region of the 16S rRNA gene in bacteria (PE16S_V4_U515_F, 5′-*ACACG ACGCT CTTCC GATCT* YRYRG TGCCA GCMGC CGCGG TAA-3′; PE16S_V4_E786_R, 5′-*CGGCA TTCCT GCTGA ACCGC TCTTC CGATC T*GGAC TACHV GGGTW TCTAA T-3′)[38] and the V9 variable region of the 18S rRNA gene in Eukarya (Il_18S_V9_1391F, 5′-*ACACG ACGCT CTTCC GATCT* YRYRG TACAC ACCGC CCGTC-3′; Il_18s_-V9_EukB, 5′-*CGGCA TTCCT GCTGA ACCGC TCTTC CGATC T*TGAT CCTTC TGCAG GTTCA CCTAC-3′)[39]. These primers also include a partial Illumina adapter sequences at the 5′ end (italicized above) and a YRYR sequence was added in the forward primer as a complexity region to help the image-processing software detect distinct clusters during Illumina sequencing. In the second step, we incorporated the full Illumina adapter and a sample-specific 9 bp barcode for library identification with overlapping primers (PE-III-PCR-F, 5′-AATGA TACGG CGACC ACCGA GATCT ACACT CTTTC CCTAC ACGAC GCTCT TCCGA TCT-3′; PE-III-PCR-001-096, 5′-CAAGC AGAAG ACGGC ATACG AGATN NNNNN NNNCG GTCTC GGCAT TCCTG CTGAA CCGCT CTTCC GATCT-3′, where Ns stand for the sample-specific barcode listed in Supplementary Table 2).

To normalize template concentrations and avoid artifacts due to overamplification of templates, we first quantified 16S rRNA gene copy numbers in each sample using real time PCR with the primers used in the first amplification (first-step PCR). This real time PCR was carried out in a final volume of 25 µL containing 5 µL of 5x HF buffer, 200 µM of dNTPs, 0.3 µM of each primer (PE16S_V4_U515_F/PE16S_V4_E786_R or Il_18S_V9_1391F/Il_18S_V9_EukB), 0.5x SYBR Green I nucleic acid stain (Invitrogen™), 2.5 U of Phusion® High-Fidelity DNA Polymerase (New England BioLabs Inc.), and 20 ng of template DNA. The amplification program consisted of an initial denaturing step at 98 °C for 3 min followed by an amplification step of 45 cycles of 30 s at 98 °C, 30 s at 52 °C, and 30 s at 72 °C, and a final extension of 5 min at 72 °C. Using the threshold cycle ($C_t$) obtained in the real time PCR, i.e., the cycle number at which template accumulation entered the exponential phase, we normalized all template concentrations for subsequent library construction to match the most dilute sample. The first-step PCR was then carried out with normalized template concentration in quadruplicates. The conditions were the same as in the real time PCR used to determine the $C_t$ value but the SYBR Green I nucleic acid stain (Invitrogen™) was excluded and the number of amplification cycles were adjusted to an appropriate $C_t$ value for the normalized templates (this was 15 and 20 cycles for 16S and 18S rRNA gene amplification, respectively). The PCR to add barcodes consisted of 9 cycles.

To purify the first-step PCR products, the quadruplicates were pooled and purified by solid-phase reversible immobilization (SPRI) paramagnetic bead technology (AgenCourt® AMPure® XP, Beckman Coulter), which is well suited for high-throughput purification of PCR amplicons. The pooled PCR reaction volumes (100 µL) were mixed with 85.5 µL of beads and incubated for 13 min to bind the DNA to the beads. Subsequently, the sample was incubated for 15 min in a magnetic rack (SPRIplate® 96-Ring) and washed with 100 µL of 70% ethanol. After drying, the DNA was incubated with 40 µL of EB buffer (Qiagen, USA) for 7 min to elute DNA followed by further 15 min incubation on the magnet to separate the DNA-containing solution from the magnetic beads.

The second-step PCR was carried out in a final volume of 25 µL containing 5 µL of 5x HF buffer, 200 µM of dNTPs, 0.4 µM of the primers PE-III-PCR-F and PE-III-PCR-01-096, 2.5 U of Phusion® High-Fidelity DNA Polymerase (New England BioLabs Inc.), and 4 µL of the purified first-step PCR as template. The amplification program consisted of an initial denaturing step at 98 °C for 2 min followed by an amplification step of 9 cycles of 30 s at 98 °C, 9 s at 70 °C, and 30 s at 72 °C, and a final extension of 2 min at 72 °C. This PCR was also done in quadruplicates, pooled after amplification, and purified by AgenCourt® AMPure® XP magnetic beads as described above.

Ribosomal RNA gene libraries were multiplexed in groups of 96 samples. Multiplexing ratios were estimated by real time PCR with Illumina sequencing primers. This PCR was carried out in a final volume of 25 µl containing 12.5 µL of 2× QuantiTec® SYBR® Green PCR kit mastermix (Qiagen), 0.2 µM of the primers PE-seq-F (5′-ACACT CTTTC CCTAC ACGAC GCTCT TCCGA TCT-3′) and PE-seq-R (5′-CGGTC TCGGC ATTCC TGCTG AACCG CTCTT CCGAT CT-3′), and 5 µl of each library sample as template. The amplification program consisted of an initial denaturing step at 95 °C for 15 min followed by an amplification step of 45 cycles of 10 s at 95 °C, 20 s at 60 °C, and 30s at 72 °C, and a final extension of 5 min at 72 °C. Multiplexed libraries (batches of 96 samples) were submitted for Illumina paired-end sequencing at the Biomicro Center (MIT, Cambridge, MA) using the MiSeq and HiSeq platforms as specified in Supplementary Table 3. The 16S rRNA gene sequencing was done by paired-end sequencing of 100 bp each read, whereas 18S rRNA gene sequencing was done by paired-end sequencing of 150 bp each read.

**Operational taxonomic unit calling**. To identify populations at high taxonomic resolution, we used the distribution-based clustering algorithm to group rRNA gene sequences into OTUs[17]. This approach takes into account both genetic

distance and the distribution of sequences across samples. Specifically, it merges sequences into a single dbc-OTU if, within an initially defined specific sequence similarity threshold (95%), they have statistically indistinguishable distribution. Sequence clusters that follow different dynamics are assigned independent OTU status even if they fall within the 95% similarity threshold. The purpose is to differentiate variation within ecologically cohesive populations (arising from sequence variation and errors as well as operon-level differences) from variation in different populations. Hence this method can result in OTUs comprising sequences of varying but usually high similarity. The method was implemented as described in the raw fastq sequence pre-processing and distribution-based clustering documentation available online (https://github.com/spacocha/Distribution-based-clustering).

We first pre-processed the raw fastq sequences output from the Illumina platform. Sequences were quality filtered with split_libraries_fastq.py from QIIME 1.3[40], using a 23 phred quality score threshold for quality filtering and retaining only sequences at least 99 bases long out of the 100 sequenced bases after quality filtering. Subsequently, primers were trimmed with trim.seqs from the Mothur 1.31 package[41] and progressively clustered into 90% identity clusters with USEARCH 5.8[42].

After sequence pre-processing distribution-based clustering was run in parallel. In the case of 16S rRNA gene sequences, only the forward read was considered due to the lack of overlap between the paired-end reads ($2 \times 100$ bp reads), whereas in the case of 18S rRNA gene sequences pair-end reads ($2 \times 150$ bp reads) were overlapped and trimmed to 120 bp long sequences prior to the progressive clustering.

Final OTUs used for this study are presented in Supplementary Data 1–6, where Supplementary Data 1 and 2 show the OTU read counts per sample for bacteria and eukaryotes, respectively. Supplementary Data 3 and 4 show the relative abundance of each OTU per sample and representative OTU sequences can be found in Supplementary Data 5 and 6 for bacteria and eukaryotes, respectively.

**OTU taxonomy assignment**. To assign each OTU to the lowest possible taxonomic category, we used the RDP classifier algorithm[43] through the QIIME 1.3 toolkit[40]. We used the 12_10 Greengenes 97% reference OTU collection (http://greengenes.secondgenome.com/downloads/database/12_10) and the eukaryotic Silva 111 reference OTU collection (http://qiime.org/home_static/dataFiles.html) for bacterial and eukaryal OTU taxonomy assignment, respectively. For each possible taxonomic level, a 0.80 confidence threshold was used to assign a specific taxon to an OTU's representative sequence. Additionally, due to a high proportion of taxonomically unassigned OTUs among eukaryotic sequences, we additionally used the BLAST algorithm included in QIIME 1.3[40] to approximate their taxonomy. Each OTU sequence was assigned the taxonomy of the best BLAST hit with a maximum $E$-value of 0.001.

OTUs classified as archaea or chloroplasts were excluded from further analysis. OTUs assigned to archaea represented an average of 0.01% while those identified as chloroplast represented an average of 7%; however, they were typically at very low proportions except on specific days when they could reach up to 30% of the sequences, presumably due to algal blooms as they were mainly placed within the Bacillariophyta (diatoms) and thus captured in our 18S rRNA gene-based OTUs.

For further analysis, we used the average for each OTU identified in the triplicate daily samples. Across the time series, triplicates from the same day were always more similar to each other than to any other samples. Supplementary Fig. 16 shows the average Jensen–Shannon square root distance between samples at different day lags. Those with zero lag (i.e., the replicates) have the smallest distance while the biggest increase in distance occurs from lag zero to lag one.

**Beta diversity estimation**. To analyze compositional change during the time series, beta diversity was estimated by computing the Jensen–Shannon distance, i.e., the square root of the Jensen–Shannon divergence, and similarity estimated as 1-Jensen–Shannon distance. We computed the Jensen–Shannon distance with the PySurvey 0.1.2 python package (https://bitbucket.org/yonatanf/pysurvey). Subsequently, we averaged distances at different time lags up to 40 days time lag for every possible time window.

**Wavelet-based identification of pairwise associations**. We are interested in identifying OTUs that are negatively or positively associated with one another, while also being co-responsive to some environmental variable. If environmental and organismal interactions occur at different frequencies, then we can identify associated pairs as those that simultaneously positively correlate at one frequency, while negatively or positively correlating at another. For example, overall physical and chemical regimes are expected to prevail over longer periods suggesting that pairs of organisms responding by growth to these conditions are positively correlated over longer periods (lower frequencies). Moreover, such growth (e.g., by characteristic primary producers) may lead to longer lasting interactions. Conversely, direct organismic interactions such as cooperation or competition and predation likely act on shorter time scales leading to positive and negative correlations at higher frequency. However, the characteristic frequency for each type of interaction cannot be known a priori and must be estimated.

To compare OTU similarities across frequencies, we perform a wavelet decomposition for each OTU time series. These analyses produce representations of the temporal data at multiple scales. At the lowest resolution of the decomposition the original signal is represented with relatively few points, and best captures low-frequency information. Similarly, at higher resolution levels of the decomposition, the signal best captures high frequency information. We can decompose the original signal for each OTU into multiple resolutions, calculate the cosine similarity for each pair of OTUs at each resolution, and then look for pairs with similarity at one resolution and similarity (positive interaction) or anti-similarity (negative interaction) at another resolution. We refer to such OTU pairs as frequency interacting.

The relevant low and high frequencies for a frequency interacting pair could change from pair to pair, depending on the nature of their interaction. For example, the OTUs could exhibit exceptionally fast or slow doubling times, and one environmental niche might change size with much more rapid dynamics than another. Therefore, for each pair of OTUs, we consider all possible low and high frequency combinations, subject to sampling duration and frequency. For each high/low-frequency combination, scores are calculated as the geometric mean of similarity and anti-similarity:

$$\text{Score}_{\text{high}^+\text{low}^+} = \sqrt{\text{similarity}_{\text{high}} \times \text{similarity}_{\text{low}}}; \text{if similarity}_{\text{high}}, \text{similarity}_{\text{low}} > 0.7$$

$$\text{Score}_{\text{high}^+\text{low}^+} = 0; \text{otherwise.}$$

(1)

$$\text{Score}_{\text{high}^-\text{low}^+} = \sqrt{-\text{similarity}_{\text{high}} \times \text{similarity}_{\text{low}}}; \text{if similarity}_{\text{high}}$$

(2)

$$< -0.5, \text{similarity}_{\text{low}} > 0.7 \text{Score}_{\text{high}^-\text{low}^+} = 0; \text{otherwise.}$$

(Note that we threshold positive similarity scores <70%, and negative similarity scores less than 50%.)

In addition, we apply a "penalty" for $\text{high}^+\text{low}^+$ scores when the high and low frequencies are very close. This is meant to account for a discretization of frequency levels in the decomposition. For instance, if a pair of interacting OTUs is driven at a "true" frequency of 3 days, and our decomposition includes a 2-day period and a 4-day period, then we might observe positive correlation at both the "high" frequency of 2 days and the "low" frequency of 4 days. However, we would not expect the "true" correlation at a 3-day period to have as strong of an effect on correlations at 32 days. Therefore, we give a higher score when the levels are more separated:

$$\text{Weighted Score}_{\text{high}^+\text{low}^+} = (k(\text{high} - \text{low}) + 1)\text{score}_{\text{high}^+\text{low}^+}.$$

(3)

For $\text{high}^-\text{low}^+$ scores we take the opposite approach; interactions receive higher weights when the frequency levels are very close. As before, we note that positive (or negative) correlation at some "true" frequency will affect correlations at the nearest levels of the decomposition. The observation of correlations of opposite signs at neighboring frequency levels is therefore less likely to represent discretization noise, and more likely to reflect a tight coupling of OTUs. Thus, we adjust $\text{high}^-\text{low}^+$ scores as follows:

$$\text{Weighted Score}_{\text{high}^-\text{low}^+} = \frac{\frac{1}{\text{high}-\text{low}}}{k + \frac{1}{\text{high}-\text{low}}}\text{score}_{\text{high}^-\text{low}^+}.$$

(4)

The parameter $k$ can be adjusted to alter the sensitivity of the overall score to this weighting, and was chosen as 0.25 for these studies. This choice corresponds to a weight of ½ when the difference between periods is 4 days. As the different levels of our wavelet decomposition approximately correspond to periods of 2, 4, 8, and 16 days, this parameter value gives relatively low weights to scores between the two highest frequencies (corresponding to periods of 2 and 4 days), and relatively high weights to scores between the lowest and highest frequencies. In studies with a different sampling period and duration, the parameter can be adjusted to achieve a similar effect.

Finally, the interaction score for a pair is taken to be the maximum of weighted scores for each frequency combination:

$$\text{Interaction Score}^{-/+} = \max_{\text{high}^-\text{low}^+} \text{Weighted Score}_{\text{high}^-\text{low}^+},$$

(5)

$$\text{Interaction Score}^{+/+} = \max_{\text{high}^+\text{low}^+} \text{Weighted Score}_{\text{high}^+\text{low}^+}.$$

(6)

Wavelet analysis was done in Python using the PyWavelets package. Multilevel decompositions were done using a "sym2" mother wavelet, and the maximum number of levels given the number of samples. For a given level of the decomposition, the similarity between two OTUs (e.g., $\text{similarity}_{\text{high}}$) was calculated as the cosine similarity of their decompositions.

**Markov clustering of frequency interacting pairs**. We next consider all frequency interacting pairs as neighbors in a graph (with separate graphs for positive and negative interactions). Edge weights in the graph are set by the interaction score between a pair of OTUs. We then use MCL (Markov Clustering Algorithm[44]) to identify OTU clusters. Briefly, MCL aims to identify local, densely connected neighborhoods in a large graph by iterating over two phases: expansion and inflation. In the initial expansion phase two-step transition probabilities are calculated by first normalizing the graph adjacency matrix to represent one-step transition probabilities from one node to any other. Next, the transition matrix is multiplied by itself to realize the two-step probabilities. In the inflation phase each column of the two-step transition matrix is normalized by an $L_p$-norm, where the inflation parameter, $p$, is chosen by the user. With increasing values of $p$, this has the effect of eliminating more low-probability transitions, and highlighting high-probability transitions.

In practice the inflation parameter can be used to explore a range of clustering granularity, as small inflation values produce a smaller number of larger clusters, while large inflation values produce a larger number of smaller clusters. We clustered our data multiple times with inflation values of 1.6, 2.0, 2.4, 2.8, 3.2, 3.6, and 4.2 (Supplementary Table 4), which are within the normal range suggested in the MCL documentation. For both $\text{high}^-\text{low}^+$ scores and $\text{high}^+\text{low}^+$ scores we found over 99% of OTUs clustered with inflation parameter up to 2.4, with a drop in OTU inclusion above this level. Clusters with this inflation value (2.4) were used in all subsequent analyses.

We used adjusted mutual information (AMI) to quantify the similarity of clusters produced in different ways (e.g., using positive vs. negative interactions). The Mutual Information between two sets of clusters is a measure of their similarity. It is calculated as:

$$\text{MI}(U, V) = \sum_i \sum_j P(i, j) \log\left(\frac{P(i, j)}{P(i)P'(j)}\right),$$

(7)

where $P(i) = \frac{|U_i|}{N}$ is the probability of being in cluster $i$ in set $U$, $P'(j) = \frac{|V_j|}{N}$ is the probability of being in cluster $j$ in set $V$, and $P(i, j) = \frac{|U_i \cap V_j|}{N}$ is the probability of being in cluster $i$ in set $U$ and cluster $j$ in set $V$. It is close to 1 when the clusters are nearly identical, and close to 0 when the clusters are highly dissimilar. AMI makes an adjustment to this score to account for the fact that the mutual information is generally higher for two clusterings with a large number of clusters. This score was implemented in Python using the function in the sklearn package.

**Simulations to validate WaveClust**. Time series were simulated to capture patterns of blooms (occurring at low frequency and higher abundance) in the midst of basal fluctuations (occurring at high frequency and lower abundance) with the addition of random noise. Specifically, we generate a random time series as $f(x) = e^{T(\sin x + b\sin\frac{x}{p} + \epsilon)}$, where $b$, $p$, and $\epsilon$ can be varied to control the bloom size, bloom period and random noise, respectively, and $T(y)$ is a threshold function $T = y, \text{if } y > 1$. For a given bloom period, we allowed for multiple blooms (with $T = 0, \text{otherwise}$. the total number determined by the length of the time series divided by the period), and a single (non-recurrent) bloom (with a random position in the time series). These models capture the qualitative features of interest that we observe in real time series (Supplementary Figs. 1A and 2A). The noise for each time series was generated as a random Gaussian variable with signal-to-noise ratio, $\frac{\|f(t)\|}{\|\epsilon\|}$, of 1.

For each of 500 random trials we simulated 8 time series with random noise and predetermined period, phase, and amplitudes (Supplementary Fig. 1A shows the eight series generated by one such trial). In each random trial, we calculated all pairwise correlations using Pearson correlation, WaveClust(+/+), and WaveClust (+/−). We then averaged these similarity matrices across all 500 trials to generate the heatmaps shown in Supplementary Fig. 1B, C, and D for each similarity metric.

To perform a sensitivity analysis of WaveClust, we varied the bloom size (b, from 2 to 20) and bloom period (p, from 10 to 50) across random trials, and calculated positive (Supplementary Fig. 2B) and negative (Supplementary Fig. 2C) association scores. In each random trial, simulated OTUs had "type 1" associations (simulated with positive correlation at high frequency and low frequency), "type 2" associations (simulated with negative correlations at high frequency and positive correlations at low frequency), or "type 3" associations (simulated with positive correlation at high frequency, but no correlation at low frequency). In Supplementary Fig. 2B and C, scores between pairs of OTUs are grouped by these three types of simulated associations.

**Microbial community predictions based on environmental data**. We used Granger causality[45] to determine the environmental drivers of each community. For each cluster, we calculated the average relative abundance of OTUs in the

cluster at each time point, and then asked which environmental variables were significantly predictive, testing for lagged values of the average abundance. Although the environmental time series were largely stationary, the average abundance of each cluster typically was not, and so we applied the method of Tado and Yamamoto[46] to calculate causality. Briefly, if either time series is non-stationary, then the causality statistic does not follow its usual ($\chi^2$) distribution. The method corrects for this by including additional lags, up to the order of integration of the non-stationary series, in the auto-regression model. Significant environmental drivers of each cluster average were reported when the $p$-value of causality passed a false discovery rate of 10%.

**Analysis of partitioning of OTUs into different communities.** In this study we have defined OTUs by the distribution-based algorithm, which allows for OTUs containing sequences of varying similarity (see above). To analyze whether closely related OTUs occur predominantly within the same or distinct communities, we first reclustered representative sequences of each dbc-OTU at 97% sequence similarity and then determined the number of OTU pairs that occurred in the same or different communities within a specific range of genetic distances. This was done by first sorting the OTU sequences by abundance (from highest to lowest relative abundance), and then clustering them by USEARCH 5.2[42] using a 97% similarity threshold. Then, we determined the co-occurrence patterns of closely related sequences within 97% OTU clusters. To this end, we computed a histogram in R of the genetic distances within each 97% OTU cluster with the function dist.dna (default parameters) within the ape R package[47], on previously aligned sequences with clustal X 2.1[48]. The vast majority of microbial diversity studies define OTUs solely based on a 97% similarity criterion, but by this approach, we could show that the majority of the interacting pairs within the 97% threshold fell into different communities. To ensure that this observation was not the result of a random distribution of close related associated pairs we tested by a $\chi^2$ goodness of fit the significance of a random community membership assignment.

**Code availability.** Software used in WaveClust analysis is publically available on github at https://github.com/brian-cleary/WaveletCombinatorics.

**Data availability.** The sequence data generated during this study have been deposited in the NCBI Sequence Read Archive repository (https://www.ncbi.nlm.nih.gov/sra) under accession numbers SRR5175890 to SRR5175997 and SRR5176042 to SRR5176255 for bacterial sequences and SRR5177223 to SRR5177507 for eukaryotic sequences. Additional data supporting the findings of this study are available as Supplementary Data 1–7.

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

## Acknowledgements

We thank Michael Cutler for extensive logistical support; Alison Takemura for processing nutrient samples; Tara Soni, Hong Xue, and Hanan Karam for field and lab support; and Gitta Szabo, Robert Ratzlaff, Otto Cordero, Nisha Vahora, Aidong Ruan, and Hans Wildschutte for assistance with sampling. We also thank Luke Miller for sharing temperature data, and Valery Kosnyrev for preparing the satellite images. We thank the NASA Ocean Ecology Laboratory, Ocean Biology Processing Group at the Goddard Space Flight Center for the MODIS-A Level 2 chlorophyll *a* and sea surface temperature data. This work was supported by grants from the U.S. National Science Foundation (OCE-1441943) to M.F.P. and the U.S. Department of Energy (DE-SC0008743) to M.F.P. and E.J.A. A.M.M.-P. was partially supported by the Ramon Areces foundation through a postdoctoral fellowship. D.J.M. was supported by the U.S. National Science Foundation (OCE-1314642) and National Institute of Environmental Health Sciences (1P01ES021923-01) through the Woods Hole Center for Oceans and Human Health.

## Author contributions

A.M.M.-P., B.C., E.J.A., and M.F.P. designed the experiments and analysis pipeline. A.M.M.-P. and B.C. carried out the WaveClust analysis. K.K. was responsible for sampling and carried out analysis of metadata. S.P.P. provided high-throughput OTU calling. D.J.M. analyzed satellite data. All authors contributed to writing and editing of the manuscript.

## Additional information

**Competing interests:** The authors declare no competing financial interests.

