## [Peer Review File · Nature Communications]

Reviewers' comments:

Reviewer #1 (Remarks to the Author):

Summary

This manuscript describes a new approach to examining microbial community time series data, using a coastal ocean time series as a development/test case. The method depends on wavelet analysis to cluster groups of organisms and identify potential interactions (both positive and negative). The frequency decomposition is an interesting and compelling approach, but I have some concerns about how the subsequent analyses were carried out. The dataset they analyzed is an exciting one, but also challenging to interpret. The authors have done a good job of extracting an interesting story from it. However, I think much remains to be done, to convince readers that the story holds water.

Major comments

Some of the methods need attention. It is not clear how replication was handled at the level of amplicon data analysis. The authors indicate that they collected triplicate water pulls from the same location during the first part of the time series, and from three spatially separated stations later in the time series. Does this mean the later part of the time series had 9 samples (3 reps per station)? The authors note "no significant differences in community structure were detectable between the two sampling methods" but this statement needs evidence to support, even if it ends up being relegated to supplementary text. What kinds of differences did they look for? Furthermore, how were replicates handled in the data analysis stage? I might have missed a detail somewhere, but it needs to be more clear. I am asking mainly because the authors focus so much on day-to-day fluctuations and I want to be convinced that variation among biological replicates (and certainly technical replicates) are much less than this variation.

The methods regarding how sequencing was conducted need more details. Was the sequencing done on an Illumina HiSeq or MiSeq? And what target read length? How were the groups of 96 samples multiplexed? Were there 96 in one lane of HiSeq, or 96 in one run of MiSeq, or something else? At first I was confused about why they used only the forward read for bacterial community data analysis, since they should have been able to overlap the paired reads (thus, I worried about the quality of the sequencing run overall, if so few paired ends would overlap). But I suppose if they were running MiSeq with 2x100, most of them would not overlap and thus this approach is reasonable.

Regarding the development of waveclust algorithm: I like the idea that environmental and biotic interactions occur on different frequencies. However, the degree of autocorrelation at these scales could strongly influence results. Fig 1D demonstrates this differential autocorrelation clearly. For example, it might be a reasonable assumption that the population of *Anabaena* is independent at a scale of 1 month. However, population dynamics will be very highly autocorrelated at the daily scale. Thus, on the daily scale, the rate of population change would be more informative than absolute population size. The difference in autocorrelation could easily inflate the strength of correlations identified. How does wavelet analysis handle this issue?

Is there a sensitivity analysis of waveclust? What influences the results? How do skewness and blooming dynamics (peak bloom size, number of blooms) influence results? Is there any way to do a validation by predicting abundance peaks of OTUs using the information from other OTUs in the identified community? Would be useful to investigate whether 1) they can qualitatively forecast an increase in abundance and 2) get a rough estimate of how much of the variance in populations can be resolved using this method.

Need to be more clear about the method for inferring a "community" from the results of waveclust. How were taxa assigned to communities?

How much does autocorrelation influence the results? For instance, if correlations were calculated after first differencing the data, what are the effects? This would indicate whether changes to populations are synchronous, rather than population size. This would be particularly interesting for the

phytoplankton, as individuals from the same generation could easily be captured in multiple samples (i.e. sampling frequency is higher than generation time frequency). It makes me apprehensive that they studied groups of taxa that peaked in abundance together. Abundance peaks are important, but also prone to higher measurement error. Such peaks also have huge influence on pairwise correlations. Furthermore, most communities appear in only one main bloom during the season. The exceptions seem to be A and B, which reappear after the upwelling disturbance. In the absence of reproducible assemblages, it might be a stretch to call these correlated taxa "communities." For most, an equivalent description would be "collections of simultaneously blooming taxa."

Lines 166-172 are a bit of a stretch. If they're going to make this conclusion (positive correlations selecting for persistent taxa, and fine scale interactions occurring within communities), they need to be more specific about a proposed mechanism to link correlation and biotic interactions, what assumptions are necessary, and citations to ecological literature.

Again, in the discussion, need to include what assumptions are involved in inferring interaction from correlation. The interchangeable usage of "correlation" and "interaction" is incorrect.

I was particularly confused about the methods section starting line 628 and corresponding results in Figure 4. When the OTUs were re-clustered, what was the actual sequence used to do the clustering? Was it a representative sequence from the 97%-OTU? Or were all sequences used? Please show the data mentioned in lines 635-638. I do not understand what the green bars are in Figure 4. Generally need more/better explanation about this part of the study.

Minor comments

The intro is vague on the theory associated with community assembly. It spends a lot of time talking about what other studies have done, but not on reasons communities would be expected to form or reviewing how biotic interactions contribute to community assembly. They later reference that biotic interactions and correlations are "expected" in ecological communities (lines 141-142), but give little background on how/why.

They cite some SPOT research, but also omit some recent studies that propose the importance of biotic interactions. Another SPOT paper (that they do cite) shows similar results to theirs, demonstrating that the dominant bacterial OTU in the ocean can change rapidly throughout the season. An alternate, more simplistic interpretation of the present study could be that the "communities" identified are really just taxa associated with the succession of dominant taxa (this relates to a major comment below).

In the abstract and a few places elsewhere, the authors refer to "plankton" as a singular noun. For example: "... microbial plankton is organized into clearly defined.." I understand that by some conventions this is accurate, but it sounds awkward. I would prefer to treat "plankton" in this context as plural.

lines 128-130: more detail about simulations would be helpful. For instance, being more specific about the kind of pattern in the input data that was recovered with the analysis.

lines 160-162: it's a poor assumption that negative correlation = negative interaction. Two competing taxa (say, two species of the same genus) often have niche overlap and will respond synchronously to changing environmental drivers.

in the section "Fine scale taxonomic differentiation of communities," some of the discussion of speciation and niche partitioning seems like an over-reach without genomic/metagenomic data. It is well known that 16S is not a sufficiently variable locus/marker to be able to track coherent populations, per se.

line 214: what do they mean by "different ecological conditions"? Environmental conditions? This section shows pretty clearly that environmental drivers can induce new community formation. For instance, it seems that upwelling could cause mass dispersal of many OTUs, thereby causing the simultaneous abundance peaks. How are these environmental drivers separated from other (presumably biotic) correlations in the waveclust analysis? Maybe this is handled in the materials and

methods.

Generally speaking, parts of the results section are speculative and could be moved to the discussion, which is short.

Materials and methods, line 313: how were replicates handled? Averaged? If not averaged, how does different replicate number influence the analysis?

Switch order of paragraphs starting on line 318 and line 322

Line 500-501. What replicates are these? The full biological replicates? See comment above too.

Line 524: co-occurrence is not an interaction.

Does the skewness of the simulated data match the skewness of the observed data? Do the simulations reproduce the bloom dynamics apparent in the real data? I am very curious about how the bloom dynamics contribute to the observed results, and wonder whether this feature was adequately reproduced in the simulated data.

Reviewer #2 (Remarks to the Author):

This is a very interesting paper that describes the detection of cohesive microbial communities in coastal waters, by using a daily timeseries of marker-gene amplicon datasets, and high-resolution OTU detection. The microbial communities turn over rapidly, and in some cases the authors provide tentative explanations for these turnovers based on environmental observations (an upwelling event, a passing hurricane).

The message is of potential high interest to marine microbiologists, and because of its use of innovative methodology, to microbial ecologists in general.

There are a few things the authors should resolve and/or clarify for the paper to be completely convincing, and yield the potential impact that it could.

First, the reproducibility of the Methods. Much more data has to be shown for the results to be reproducible for others. This is an even more important point due to the exemplary function this paper may have on others in the community. People will want to reproduce the results presented here, before trying similar approach on their own environment of interest. Not only the SRA identifiers should be given, but also all relevant intermediate and final results in the form of Additional Files/Supplementary Tables. Some detailed points are listed below.

Second, the statistics need to be carefully reviewed. Due to the novelty of the approach, it is difficult to assess the significance of the observation - i.e. the rapid, concerted emergence and decline of cohesive microbial communities. It is unclear to me whether this is significant - the authors should show, that with randomly permuted timeseries for each OTU such clusters do not appear. (How many randomizations are doable defines a lower bound to the P-value of the study.) This could be important since the OTUs are defined at very different levels - first at 90% using UClust and then at 95% using Distribution-based clustering, yet the majority of all pairs at 97%-99% identity fall into different communities (l.198) that turnover very rapidly (l.142). Including time series permutations may answer the question whether all this is significant. While the authors show that their method works on "ideal" simulated time series (l.610 and Suppl Fig 2) that is very different than showing it does NOT work for shuffled real time series. (I.e. they could still be describing noise.)

Additional specific points:

- It would be great to see a lot more detail about the reconstructed communities: How many communities were there? How many members per community? How many sequences were not included in any community? What were the sequences (supplementary Fasta file)? Add a plot showing the within- and between-community sequence identities (with standard deviations), same for the OTUs - this should also reveal the extent of overlaps between communities at different sequence

identity cutoffs.

- It is weird to have a figure without a legend - please move Suppl Fig 1 to Fig 1 and remove from supplement. Also Fig 3B and Suppl Fig 3B are identical.
- L.42 replace "constrained" with "studied"?
- L.145. What is the unit of the Y-axis of Fig 3A? If relative contribution, then why does it go >1 , and similarly why does it go <1 ? (I would expect a figure more like Fig 1A/B.)
- L.157. The mutual information score is not described in the Methods section, and it is unclear if 52% is high or low.
- L.169-172. Negative correlations over longer periods were not assessed.
- L.201 "frequent speciation": this is confusing since sequences were clustered at 95% nucleotide identity, yet 97%-99% ID sequences tended to be in different clusters. So what does speciation mean here? Could you discuss why it would (not) make sense to define more stringent OTU cutoffs?
- L.233: mention community D.
- L.236. How can I see "the passage of a hurricane system and followed by gradual, seasonal cooling of the water" in Fig. 3C and Suppl Fig 3C?
- L.306 Could the geographic coordinates be given in global format (rather than Google maps).
- L.316-317 states "No significant differences in community structure were detectable between the two sampling methods". This is not shown, and
- The results of replicates should be shown. Although experimental variation is briefly discussed, there is no supporting information for this and no error bars are shown anywhere (e.g. L.316 states: "No significant differences in community structure were detectable between the two sampling methods" without reference to evidence - here it is unclear what "two sampling methods" you are talking about).
- L.318: "Accompanying qualitative metadata include field notes, microscopy, photographs, and videos." It is unclear how this is used: if it is used then explain how, and please provide the relevant files for reproducibility. If it is not used, then this statement can be removed.
- L.319. Please include the raw data files of all the environmental parameters in a supplement.
- L.322: "each day as triplicate" - in line 314 it says only "from Day 259 forward replicates for each sample type were collected"?
- L.323: Can you give a rough estimate of the time until return to the lab?
- L.327: "collected daily and processed and analyzed" - now it looks like "daily" is only about "collected", which is to be expected. Please indicate if they were also "processed and analyzed" daily, at the end of the experiment, or in batches at some different frequency.
- L.382. The potential effects of 3 rounds of PCR cycles should be discussed: the first round of PCR had an unknown number of cycles, the second had nine cycles, and the third had 45 cycles). I'm thinking of primer biases, skewing abundances, generation of chimeric sequences.
- L.419. Please provide a table with Ct used.
- L.450. Statistics of the sequencing runs and sequencing QC should be provided.
- L.568: "roughly centers the dynamic range of weighting around the frequencies we were able to capture given our sampling period and duration." - unclear what this means and what it is based on?
- L.596: "we found over 99% of OTUs clustered with inflation parameter up to 2.4, with a drop in OTU inclusion above this level" - please show this.
- L.623. Could you add a brief nontrivial insight about "the method of Tado and Yamamoto"?
- L.624. For the Granger causality of the environmental drivers of a cluster, associations were considered significant at a p-value <0.05 . However, as far as I can tell this was not corrected for multiple testing while it is unclear how many comparisons were made.
- L.786: "Edge width scales with p-values" - all the edges have the same width?
- Figure 4 is confusing - it is unclear how to see the differences discussed in the text. Maybe rescale everything to 100% or show observed/expected ratios? Also I'm very interested to see if the suggested permutation analysis (see my comment above) will show that the observation is significant: that at 97%-99% nucleotide identity closely related sequences are more frequently in different

clusters than expected. I think that this is a different question than the one the authors answer with their "random community membership assignment" analysis (l.797) - but please explain this in detail in the Methods (I could not find it).

Reviewer #3 (Remarks to the Author):

Martin-Planetero et al Review.

Main comments:

Intensive time-series datasets and analysis of the combined effects of broad environmental changes with biotic interactions in the microbiome are key areas of research.

Martin-Platero and colleagues combine a novel wavelet-based community analysis method with intensive multi-marker coastal plankton time series. The results are consistent with past work, but also reveal surprisingly sharp transitions between community types that continue unabated throughout the sampling period. Thus, if you don't like the microbes present on one day, wait for the next! This observation, like Toynbee's criticism of history teaching as 'one damned fact after another', could easily devolve into meaninglessness, and I was initially afraid the manuscript would simply compile algorithmic results and high rates of community change and say 'good enough'.

Instead, the authors present a thoughtful and satisfactory narrative that both addresses broad questions about niche partitioning (lines 190-193), the evolutionary scales at which that niche partitioning occurs (Fig. 4), and the role of infrequent or 'special' events in shaping the planktonic community (see especially lines 222-257).

The novel methods presented (with a little additional testing, see below), and in particular the innovative way in which they are used in a real analysis seem both novel and broadly applicable to many systems. As much as any specific finding about this system, I find the authors handling of the tension between highly specific, partially stochastic events; long-term seasonal trends; and short-term high-frequency interactions to be exemplary, and of broad interest beyond the marine biology community. For example, investigators in the human microbiome must contend both with long-term processes like aging, but also inconsistencies in diet, infrequent antibiotic use, pathogen exposure, etc, etc. across individuals. Therefore, I think this integrative approach - in terms of both new methods and how they are applied and interpreted - will be of interest to a wide range of readers, and suitable in scope for the broad readership of Nature Communications.

I also note that the authors are careful in their handling of one potential criticism: that the changes observed by sampling one location repeatedly over time reflect both movement of water masses and microbial succession within those water masses. I do not view this potential criticism as a significant limitation of the paper as written because: a) it is similar and comparable to other oceanographic sampling b) for water-quality monitoring passage of different microbe-bearing waters by a particular site has practical consequences and c) the time-series captures both cases where this water-body-movement is likely important, and periods when broader seasonal trends dominate.

Major Comment: More quantification needed on the interaction of distribution-based clustering and wavelet analysis

There is currently a split in the field between sub-OTU methods that operate on a per-sample basis, and those that use information on distribution across samples. For some study questions, this split

probably doesn't matter too greatly, as broad trends in taxonomic change or beta-diversity will likely emerge regardless. For the more nuanced questions of temporal association tested here, I found myself wondering whether a flexible clustering criterion defined based on distribution is problematic.

If we define OTUs based on ecological cohesiveness, and then test a hypothesis about ecological cohesiveness over time, are we predisposing the results to favor sharp turnovers and/or correlated outcomes to a greater degree than if we used a per-sample method (perhaps deblur)?

In other words, is it possible to show (or have you already shown) that the false-positive rate for detection of cohesive communities is not increased when using distribution-based clustering? If so, can we 'subtract out' these effects from the observed trends?

Page 4 (lines 86-89).

Here's what I'm thinking: take the 'conventional' OTUs already generated for relatedness analysis (e.g. lines 632-636) or a non-distribution based sub-OTU method like deblur (in a perfect world, perhaps both). Compare these to the distribution-based OTUs. Use a permutational procedure to destroy real temporal signal and co-association in each collection of OTU tables. Then apply wavelet analysis and look for false-positive associations. In particular, I am interested in a quantification of whether there are more FPs in the distribution-based OTUs than the conventional OTUs or deblurred sequences, and whether the OTU clustering threshold has a strong effect?

I appreciated the existing simulations testing detection power for the wavelet-based approach used here vs. e.g. Pearson correlation. Regardless of the outcome, a better quantification of these interactions would strengthen the paper.

Minor comments:

Line 24. How do we feel about using 'protozoa'? Would microbial eukaryotes (e.g. including oceanic fungi) or even protists (same idea but without the connotation of primitive animals) be more appropriate?

Line 28. Do we regard plankton a countable noun? If so, consider Is are.

Line 36. 'in spite of physical instability in the coastal ocean, defined communities of organisms assemble'. This conclusion to the abstract seemed out of place to me, and a bit confusing in terms of it's actual meaning. On the one hand, 'defined communities of organisms assemble' begs the question of what the other alternative is. I think it could be sharpened in order to not push readers out before they have a chance to get to some of the neat analysis later in the paper. Setting aside the phrasing, isn't this the opposite of your conclusion? On the other hand, Based on the network analysis and hypotheses presented later, isn't it precisely the physical instability of the ocean (warm temperature water bodies; sediment-bearing upwelling events; seasonal cooling, etc.) that allows the sharp community transitions reported here to occur?

Line 99-105. Haven't these 'seed bank' dynamics also been reported in several other studies that should be cited? E.g. In the Western English Channel (Caporaso et al., 2012; <https://www.ncbi.nlm.nih.gov/pmc/articles/PMC3358019/>), and indirectly by comparison of deep sequencing of single samples to global samples from ICoMM (Gibbons 2013; (<http://www.pnas.org/content/110/12/4651.short>)). I should emphasize that I do not mention these other studies because I think they cut into the novelty of this study (which I think lies elsewhere) but merely because I think they are an important context that might help reinforce or refine the interpretations here.

Lines 190 – 193. I thought this was a really nice passage relating the specific observations described here (rapid turnover of associated communities) to broader ecological questions about diversity and niche space. I wonder if there is a way to work a short version of this idea into the abstract (perhaps in place of the last sentence?)

Lines 222 – 257 I found this final passage quite remarkable. It highlights quite effectively the intersection of oceanographic, biological, and meteorological processes that we must deal with in trying to understand real natural systems. Often, some of the factors that matter most – are messy, surprising, and perhaps not innately appealing to the simple stories we as humans like to tell (and the simple hypotheses we like to test). The combination of a newer method for community analysis, along with this thoughtful narrative of broader processes affecting the environment seems closer to a real understanding of the system than tests of any one single-factor hypothesis considered in isolation. These observations also speak to the value of intensive time-series datasets, and suggest many jumping-off points for subsequent studies.

Response to reviewers' comments.

We would like to sincerely thank all the reviewers for putting so much work into our manuscript. We feel that addressing the comments has greatly strengthened the manuscript and has made us critically evaluate many points. We greatly appreciate the effort and hope that our revisions have satisfactorily addressed the questions.

We have followed most of the comments and this has led to a greatly expanded testing of the method. Moreover, in response to questions of what environmental factors structure the communities, we have invited Dennis McGillicuddy from the Woods Hole Oceanographic Institution to join our team. Dennis is a physical oceanographer and provided satellite data to cross check whether mesoscale features of the coastal ocean correspond to community boundaries. Although frequent cloud cover prevents a complete analysis, the comparison shows that it is possible to identify transient ocean features that correspond to community boundaries identified in our work. We show the results in Supplementary Figure 13.

We have numbered the responses in order to facilitate cross-referencing between reviewer responses that address similar points, and we have added line and figure numbers to enable facile cross-checking with the text. Moreover, all changes in the text are highlighted in yellow.

Reviewer #1:

Summary

This manuscript describes a new approach to examining microbial community time series data, using a coastal ocean time series as a development/test case. The method depends on wavelet analysis to cluster groups of organisms and identify potential interactions (both positive and negative). The frequency decomposition is an interesting and compelling approach, but I have some concerns about how the subsequent analyses were carried out. The dataset they analyzed is an exciting one, but also challenging to interpret. The authors have done a good job of extracting an interesting story from it. However, I think much remains to be done, to convince readers that the story holds water.

Major comments

Some of the methods need attention. It is not clear how replication was handled at the level of amplicon data analysis. The authors indicate that they collected triplicate water pulls from the same location during the first part of the time series, and from three spatially separated stations later in the time series. Does this mean the later part of the time series had 9 samples (3 reps per station)? The authors note "no significant differences in community structure were detectable between the two sampling methods" but this statement needs evidence to support, even if it ends up being relegated to supplementary text. What kinds of differences did they look for?

(1) We now provide more detail on these issues in the section "Environmental sampling and metadata" (line 393ff). Briefly, throughout the time series, we collected 3 samples each day

where these 3 replicates were taken at the same location during the first part of the sampling program and at separate locations (~20 meters apart) during the second part.

Thank you for catching the inaccurate statement about the difference in community structure. In fact, due to the temporal component, the samples are not directly comparable. For example, across the second, spatially separate, sampling scheme, the last samples showed higher variance than the earlier ones, suggesting increased small-scale patchiness. However, this within day variation was much smaller than between consecutive days. We have changed the text to (line 396): “Independent of the sampling scheme, variation between consecutive days was always much larger than within days (Kruskal-Wallis chi-squared = 228.63, df = 87, p-value = 1.145e-14 for the entire series, and Kruskal-Wallis chi-squared = 76.722, df = 33, p-value = 2.448e-05 for the spatially separate stations).”

Furthermore, how were replicates handled in the data analysis stage? I might have missed a detail somewhere, but it needs to be more clear. I am asking mainly because the authors focus so much on day-to-day fluctuations and I want to be convinced that variation among biological replicates (and certainly technical replicates) are much less than this variation.

(2) In addition to the clarification mentioned above, we have expanded the explanation at the end of the section “OTU taxonomy assignment” (line 599) where we state that “we used the average for each OTU identified in the triplicate daily samples. Supplementary Fig. 16 shows the average Jensen-Shannon square root distance between samples at different day lags. Those with zero lag (i.e., the replicates) have the smallest distance while the biggest increase in distance occurs from lag zero to lag one.”

The methods regarding how sequencing was conducted need more details. Was the sequencing done on an Illumina HiSeq or MiSeq? And what target read length? How were the groups of 96 samples multiplexed? Were there 96 in one lane of HiSeq, or 96 in one run of MiSeq, or something else? At first I was confused about why they used only the forward read for bacterial community data analysis, since they should have been able to overlap the paired reads (thus, I worried about the quality of the sequencing run overall, if so few paired ends would overlap). But I suppose if they were running MiSeq with 2x100, most of them would not overlap and thus this approach is reasonable.

(3) We now include information on the exact platform used for samples in **Supplementary Table 3**. As the reviewer suspects, we used 100bp paired reads resulting in non-overlapping sequences in the case of 16S sequences. Hence we used the forward read. For 18S sequences we had 150 bp pair end overlapping reads.

*Regarding the development of waveclust algorithm: I like the idea that environmental and biotic interactions occur on different frequencies. However, the degree of autocorrelation at these scales could strongly influence results. Fig 1D demonstrates this differential autocorrelation clearly. For example, it might be a reasonable assumption that the population of *Anabaena* is independent at a scale of 1 month. However, population dynamics will be very highly autocorrelated at the daily scale. Thus, on the daily scale, the rate of population change would be more informative than absolute population size. The difference in autocorrelation could easily inflate the strength of correlations identified. How does wavelet analysis handle this issue?*

(4) Thank you for raising this issue. In our original manuscript, it was both unclear what the true levels of autocorrelation were, and how these should be handled by our algorithm. In our revision, we clarify (**lines 209-227**) that, while **Figure 1D** shows that population similarity (at the level of OTUs) is around 50-60% with a 1-day lag, the autocorrelation within each time series is actually much lower (median of 19%). To investigate if this level of autocorrelation has a substantial effect on our results, we show in **new analysis** that the correlation between high-frequency similarity scores calculated from the original data and high-frequency scores calculated from 1st difference data is 86% [**Supplementary Figure 7**], and that clusters generated using high-frequency scores from 1st difference data in place of the original scores are highly similar to the original clusters (81% mutual information for +/+, 82% mutual information for +/-).

Although autocorrelation on the daily scale does not seem to have a large effect on our results, the Reviewer's point that it *could* affect the results still stands. In our revised version, we point this out, and suggest that, if autocorrelation levels are high, then it may be appropriate to cluster and evaluate the differenced data (**line 223-227**).

Is there a sensitivity analysis of waveclust? What influences the results? How do skewness and blooming dynamics (peak bloom size, number of blooms) influence results? Is there any way to do a validation by predicting abundance peaks of OTUs using the information from other OTUs in the identified community? Would be useful to investigate whether 1) they can qualitatively forecast an increase in abundance and 2) get a rough estimate of how much of the variance in populations can be resolved using this method.

(5) This is a very good suggestion – we have added a sensitivity analysis in simulated data to assess how peak bloom size, number of blooms, and the period of blooms affect our results. These results are presented in **new Supplementary Figure 2** and in **line 148-156**. Briefly, time series were simulated to capture patterns of blooms (occurring at low frequency and higher abundance) in the midst of basal fluctuations (occurring at high frequency and lower abundance) with the addition of random noise. We find that positive associations (between a pair of OTUs that are correlated at both low and high frequency; **Supplementary Figure 2B**) are more difficult to detect with our method than negative associations (between OTUs that are anti-correlated at high frequency; **Supplementary Figure 2C**). For both types, our method is less sensitive with increasing bloom abundance, because the high frequency, basal fluctuations are partially masked by the high amplitude, low frequency changes. The results were largely robust to the period of the bloom. This was true whether we allowed for only one bloom per time series (as in representative examples in **Supplementary Figure 2A**), or multiple blooms. The details of these simulations are found in our revised **Methods**, and we briefly describe them below in response to the Reviewer's minor comments.

Need to be more clear about the method for inferring a "community" from the results of waveclust. How were taxa assigned to communities?

(6) Thank you for pointing this out, and apologies for leaving this unclear in the main text. We have added a clarification (**lines 141-143**) that communities are identified by graph clustering, using a combination of high- and low-frequency correlations to define edge weights.

How much does autocorrelation influence the results? For instance, if correlations were calculated after first differencing the data, what are the effects? This would indicate whether changes to populations are synchronous, rather than population size. This would be particularly interesting for the phytoplankton, as individuals from the same generation could easily be captured in multiple samples (i.e. sampling frequency is higher than generation time frequency).

(7) As discussed in response to Reviewer's major comment above, in our data autocorrelation does not have a large influence on the results, likely because the magnitude of the correlation is low, or because it decays quickly. In our revised text, we adopt the Reviewer's point, and suggest that the analysis should be performed on differenced data when the levels of autocorrelation are higher (**line 223-227**).

It makes me apprehensive that they studied groups of taxa that peaked in abundance together. Abundance peaks are important, but also prone to higher measurement error. Such peaks also have huge influence on pairwise correlations. Furthermore, most communities appear in only one main bloom during the season. The exceptions seem to be A and B, which reappear after the upwelling disturbance. In the absence of reproducible assemblages, it might be a stretch to call these correlated taxa "communities." For most, an equivalent description would be "collections of simultaneously blooming taxa."

(8) As mentioned above, our findings are robust towards using differenced data. We further believe that community is an appropriate term for the following reasons. In the revised version, we have added satellite data (**Supplementary Figure 13**). These allow in many cases identification of spatially and temporally limited events during which the communities develop, i.e., communities are features of coastal ocean dynamics and have identifiable spatial and temporal boundaries. This aspect is now discussed in the revised results **line 267-332**. A further important point is that each community contains distinct primary producers. As we discuss, these likely produce different types of high molecular weight organic matter, which is degraded by specific sets of bacterial taxa as has previously been shown in the North Sea (Teeling et al 2012). Finally when particulate organic matter forms and is colonized, fine-scale spatial interactions occur (see for example Datta et al. 2016) (**line 51-54**). Because much of this particulate organic matter is algae derived, it follows that its composition may also be community specific. It is therefore likely that many microbial taxa are interdependent and interact in these communities.

One might also argue that in microbial systems, "collections of simultaneously blooming taxa" are communities since they respond to specific sets of conditions such as carbon produced by other organisms. We also note the high specificity of the blooms, i.e., taxa only bloom in one community and not in others as would be expected if all taxa only responded to very general environmental factors.

Lines 166-172 are a bit of a stretch. If they're going to make this conclusion (positive correlations selecting for persistent taxa, and fine scale interactions occurring within communities), they need to be more specific about a proposed mechanism to link correlation and biotic interactions, what assumptions are necessary, and citations to ecological literature. Again, in the discussion, need to include what assumptions are involved in inferring interaction from correlation. The interchangeable usage of "correlation" and "interaction" is incorrect. I was particularly confused about the methods section starting line 628 and corresponding results in Figure 4. When the OTUs were re-clustered, what was the actual sequence used to do the clustering? Was it a representative sequence from the 97%-OTU? Or were all sequences used? Please show the data mentioned in lines 635-638. I do not understand what the green bars are in Figure 4. Generally need more/better explanation about this part of the study.

(9) We agree with the reviewer that correlation does not necessarily equate interaction (although the interchangeable use has become quite common in the recent literature). However, in the paragraph preceding the section mentioned by the reviewer we do exactly as the reviewer suggests: We pair the observation of correlations with interpretation based on literature by discussing that the peaks of specific primary producers are frequently correlated to peaks of specific bacterial taxa; several of these taxa have previously been shown to be degraders of algal exudates in a highly specific and reproducible manner (citations 8 and 10, **line 236**). Based on this observation, we further discuss potential implications for interpretation of microbial community structure.

The paragraph the reviewer refers to does not discuss observations based on correlations among taxa but demonstrates that closely related taxa tend to fall into different communities. To interpret this further, we look into the ecological literature where phylogenetic structure of communities has been used to suggest mechanisms of assembly. We cite a previous application to microbial communities (ref. 28) but the approach has been more common in the plant literature, which is also discussed at length in ref. 28.

To clarify the issue of how OTUs were clustered for this analysis, we have extended the description in several places. We think that much of the confusion stems from the fact that we did not explain that distribution based clustering results in clusters near 100% sequence similarity. We have added a statement in the results (**line 90**): "To maximize ecological resolution of OTUs, we used distribution-based clustering, which does not assume a fixed sequence similarity cut-off to define OTUs but instead identifies the sequence similarity at which clusters display cohesive behavior across samples (ref. 17). This approach usually yields clusters comprising very closely related sequences and resulted in". We have also extended the description also in the methods section under:

- (1) "Operational taxonomic unit (OTU) calling" (**line 551**): "This approach takes into account both genetic distance and the distribution of sequences across samples. Specifically, it merges sequences into a single OTU if, within an initially defined specific sequence similarity threshold (95%), they have statistically indistinguishable distribution. Sequence clusters that follow different dynamics are assigned independent OTU status even if they fall within the 95% similarity threshold. The

purpose is to differentiate variation within ecologically cohesive populations (arising from sequence variation and errors as well as operon-level differences) from variation in different populations. Hence this method can result in OTUs comprising sequences of varying but usually high similarity.”

- (2) "Analysis of partitioning of closely related OTUs into different communities" section (line 754). Briefly, the sequences we have clustered at a 97% similarity cut-off were representative sequences of each dbc-OTU (distribution based clustering), which are all much below the 97% threshold. We then determined sequence similarity of OTU pairs and determined at which genetic distances they occurred in the same or different communities.

Finally, we have introduced some clarification in the caption of **Figure 4**. This figure shows 3 histograms, which are overlapped. Blue and red histograms show the frequency of OTU pairs that fall in different or the same community, respectively, while green bars (at the back) show the total pair frequency (i.e. the sum of blue and red bars).

Minor comments

The intro is vague on the theory associated with community assembly. It spends a lot of time talking about what other studies have done, but not on reasons communities would be expected to form or reviewing how biotic interactions contribute to community assembly. They later reference that biotic interactions and correlations are “expected” in ecological communities (lines 141-142), but give little background on how/why.

(10) We appreciate the suggestion of adding more reasons why interactions would be expected and have attempted to do so while still preserving the length and giving due credit to past time series analyses, which have become increasingly important in hypothesizing microbial interactions. We added (line 51): “For example, direct competitive or cooperative interactions may lead to rapid micro-scale successions on suspended organic particles (Datta et al. 2016) while large-scale algal blooms may trigger growth of bacteria that degrade specific algal exudates (Teeling et al. 2012). Due to the apparent difference in scales of such interactions, it remains unknown...”

They cite some SPOT research, but also omit some recent studies that propose the importance of biotic interactions. Another SPOT paper (that they do cite) shows similar results to theirs, demonstrating that the dominant bacterial OTU in the ocean can change rapidly throughout the season. An alternate, more simplistic interpretation of the present study could be that the “communities” identified are really just taxa associated with the succession of dominant taxa (this relates to a major comment below).

(11) We agree that some taxa are associated with the succession of dominant taxa but we posit that this is an interaction expected in a community (see also our comments about dominant primary producers and their associated bacteria). After all, a forest could be regarded as a bloom of trees with associated taxa, the difference being in the much longer generation times.

In the abstract and a few places elsewhere, the authors refer to “plankton” as a singular noun. For example: “... microbial plankton is organized into clearly defined..” I understand that by some conventions this is accurate, but it sounds awkward. I would prefer to treat “plankton” in this context as plural.

(13) We have followed the reviewers suggestion.

lines 128-130: more detail about simulations would be helpful. For instance, being more specific about the kind of pattern in the input data that was recovered with the analysis.

(14) We have added more detail regarding simulations to the revised **Methods** section (**line 751**). Briefly, our time series were simulated to capture patterns of blooms (occurring at low frequency and higher abundance) in the midst of basal fluctuations (occurring at high frequency and lower abundance) with the addition of random noise. Specifically, we generate a random time series as $f(x) = e^{T(\sin x + b \sin \frac{x}{p} + \epsilon)}$, where b , p , and ϵ can be varied to control the bloom size, bloom period and random noise, respectively, and $T(y)$ is a threshold function $T = y, \text{ if } y > 1$
 $T = 0, \text{ otherwise}$. For a given bloom period, we allowed for multiple blooms (with the total number determined by the length of the time series divided by the period), and a single (non-recurrent) bloom (with a random position in the time series). These models capture the qualitative features of interest that we observe in real time series (**new Supplementary Figure 2A**).

lines 160-162: it's a poor assumption that negative correlation = negative interaction. Two competing taxa (say, two species of the same genus) often have niche overlap and will respond synchronously to changing environmental drivers.

(15) We agree with the reviewer. We have gone through the manuscript and replaced “interaction” with “association” where appropriate and have added an explicit explanation that we mean “correlated dynamics” by interactions when we introduce the waveclust approach in the results (**line 126**).

in the section “Fine scale taxonomic differentiation of communities,” some of the discussion of speciation and niche partitioning seems like an over-reach without genomic/metagenomic data. It is well known that 16S is not a sufficiently variable locus/marker to be able to track coherent populations, per se.

(16) We agree that this point should be well known but the reality is that it is not or at least frequently ignored. Also, our point explains why one might expect the most closely related OTUs to be partitioned into different communities. Hence we believe this is a valuable discussion point in the context of our observations.

line 214: what do they mean by “different ecological conditions”? Environmental conditions?

This section shows pretty clearly that environmental drivers can induce new community formation. For instance, it seems that upwelling could cause mass dispersal of many OTUs, thereby causing the simultaneous abundance peaks. How are these environmental drivers separated from other (presumably biotic) correlations in the waveclust analysis? Maybe this is handled in the materials and methods.

(17) Unfortunately, WaveClust cannot differentiate biotic from abiotic drivers. That would have to arise from appropriate metadata, which are nearly impossible to get at this point. Very likely the factors are complex combinations of biotic and abiotic factors (which we now explicitly state in the section, **line 267**). As it is currently nearly impossible to characterize difference in types of dissolved and particulate carbon, this point is nearly impossible to address.

Generally speaking, parts of the results section are speculative and could be moved to the discussion, which is short.

(18) We have considered the suggestion but we think that moving interpretation of results into the discussion might make for a very dry reading and would therefore like to keep the structure intact.

Materials and methods, line 313: how were replicates handled? Averaged? If not averaged, how does different replicate number influence the analysis?

(19) This point is addressed in the response to major comments.

Switch order of paragraphs starting on line 318 and line 322

(20) Makes sense. Done.

Line 500-501. What replicates are these? The full biological replicates? See comment above too.

(21) We hope we have adequately addressed this point in response to the point about sample replication raised earlier.

Line 524: co-occurrence is not an interaction.

(22) Removed co-occurrence.

Does the skewness of the simulated data match the skewness of the observed data? Do the simulations reproduce the bloom dynamics apparent in the real data? I am very curious about how the bloom dynamics contribute to the observed results, and wonder whether this feature was adequately reproduced in the simulated data.

(23) We interpret “skewness” here to be bloom size, number of blooms, and bloom period, as mentioned by the Reviewer’s major comment above. As mentioned in response to major and minor comments above, our simulations capture the qualitative features of low frequency blooms amidst high frequency basal fluctuations, and demonstrated in **new Supplementary Figure 2A**.

Reviewer #2:

This is a very interesting paper that describes the detection of cohesive microbial communities in coastal waters, by using a daily timeseries of marker-gene amplicon datasets, and high-resolution OTU detection. The microbial communities turn over rapidly, and in some cases the authors provide tentative explanations for these turnovers based on environmental observations (an upwelling event, a passing hurricane).

The message is of potential high interest to marine microbiologists, and because of its use of innovative methodology, to microbial ecologists in general.

There are a few things the authors should resolve and/or clarify for the paper to be completely convincing, and yield the potential impact that it could.

First, the reproducibility of the Methods. Much more data has to be shown for the results to be reproducible for others. This is an even more important point due to the exemplary function this paper may have on others in the community. People will want to reproduce the results presented here, before trying similar approach on their own environment of interest. Not only the SRA identifiers should be given, but also all relevant intermediate and final results in the form of Additional Files/Supplementary Tables. Some detailed points are listed below.

(1) We agree that it will be helpful to have additional information. We have deposited all sequences in the SRA archive and give additional supplementary tables to facilitate the reproduction of our analysis. We have added the OTU tables for bacteria and eukaryotes with both, read counts (SuppFile1_otu_table_bacteria_unfrac_Counts.xlsx and SuppFile2_otu_table_eukarya_unfrac_Counts.xlsx) and relative abundances (SuppFile3_otu_table_bacteria_unfrac_RelAbund.xlsx and SuppFile4_otu_table_eukarya_unfrac_RelAbund.xlsx). Each table contains the OTU ID as the first column, a last column with the taxonomy assignment, and sampling days in middle columns. Briefly sample names are coded as follow:

10N.XXX.YY, e.g. 10N.204.37: 10N stands for the sampling site, 204 indicates the day of the year, and 37 stands for the replicate.

Additionally we have included a fasta file with representative sequences of each dbc-OTU for both bacteria and eukaryotes (SuppFile5_representative_sequences_bacteria_unfrac.fasta and SuppFile6_representative_sequences_eukarya_unfrac.fasta)

All Supplementary Files are referenced in the text **line 574-578**.

Second, the statistics need to be carefully reviewed. Due to the novelty of the approach, it is difficult to assess the significance of the observation - i.e. the rapid, concerted emergence and decline of cohesive microbial communities. It is unclear to me whether this is significant - the authors should show, that with randomly permuted timeseries for each OTU such clusters do not appear. (How many randomizations are doable defines a lower bound to the P-value of the study.) This could be important since the OTUs are defined at very different levels - first at 90% using UClust and then at 95% using Distribution-based clustering, yet the majority of all pairs at

97%-99% identity fall into different communities (l.198) that turnover very rapidly (l.142). Including time series permutations may answer the question whether all this is significant. While the authors show that their method works on "ideal" simulated time series (l.610 and Suppl Fig 2) that is very different than showing it does NOT work for shuffled real time series. (I.e. they could still be describing noise.)

(2) We would first like to point out that OTUs are not defined at the different levels the reviewer cites and apologize for the misleading description. We use distribution-based clustering (dbc), which uses pre-clustering of sequences at 95% as a step in the computation of cohesively behaving OTUs. As detailed in the **response to reviewer #1 (point #9)**, we have now attempted to explain the procedure better in several places and we hope this addresses the reviewer's concerns.

The suggestion of permutations is a very important point, and we thank the Reviewer for raising the issue. Accordingly, we have performed **new analysis** to assess statistical significance. Specifically, we permuted each time series independently and calculated wavelet similarity scores, noting for each pair of OTUs if the scores were greater than those observed in the original data. We repeated these permutations 50,000 times, and used the results to calculate empirical p-values.

This analysis allowed us to show that the real data contain significant structure not present in shuffled time series. First, the original data have a clear enrichment for significant p-values (shown in **new Supplementary Figure 6A**). We used these p-values to identify similarity scores (in the original data) that pass a false discovery rate (FDR) of 10%, and found that clusters that use both the FDR and original thresholds are in good agreement with our original results (55% mutual information). If we repeat the FDR and clustering analysis using shuffled time series in place of the original, we observe very little clustering (just a single cluster containing only 122/9,660 OTUs). We also note that using the conservative FDR test, only 5% of the scores that passed the original thresholds also pass the FDR test. Thus, overall cluster structure remains robust, even if less stringent connections between nodes are allowed. We describe this analysis in the revised **Results** section (**line 193-207**), and note that the more conservative (and computationally intensive) method with FDR thresholds should be used to identify high-confidence associations between individual pairs of OTUs.

Additional specific points:

- It would be great to see a lot more detail about the reconstructed communities: How many communities were there? How many members per community?

(3) This information is provided in **Supplementary Table 2**.

How many sequences were not included in any community?

(4) As detailed in the first paragraph of the results, there were 49,637 OTUs and of these we picked 9,660 that occurred at least on 10 days. All of the latter were included in the WaveClust analysis (**line 93**).

What were the sequences (supplementary Fasta file)?

(5) As stated in response to Reviewer's comment above about data reproducibility, we have included two fasta files with the sequences (**Supplementary files 5 and 6**) containing representative sequences of each OTU.

Add a plot showing the within- and between-community sequence identities (with standard deviations), same for the OTUs - this should also reveal the extent of overlaps between communities at different sequence identity cutoffs.

(6) We agree that this analysis can enlighten the understanding of community assembly and we have carried out the suggested test. We find that within- and between-community sequences identities show a highly overlapped distribution, probably due to the fact that taxa are continuously present but peak during limited times. Nevertheless there is a slight but significant difference (Wilcox test p-value < 2.2e-16) with a smaller similarity between- than within-community. However, considering the specifics of the data, we think adding the plot would require substantial explanation while not substantially improving the explanation of the patterns we see.

- It is weird to have a figure without a legend - please move Suppl Fig 1 to Fig 1 and remove from supplement. Also Fig 3B and Suppl Fig 3B are identical.

(7) Agreed.

- L.42 replace "constrained" with "studied"?

(8) Agreed but we changed to "understood".

- L.145. What is the unit of the Y-axis of Fig 3A? If relative contribution, then why does it go >1, and similarly why does it go <<1? (I would expect a figure more like Fig 1A/B.)

(9) The y-axis shows the cumulative relative abundance, i.e., the sum of relative abundances of each OTU. It can be >1 because the relative abundances for bacteria and eukaryotes were calculated separately and added for each community. Therefore more diverse communities can exceed 1 while less diverse communities with lower number of OTUs can reach values <<1. The number of OTUs associated with each community is shown in **Supplementary Table 2**.

- L.157. The mutual information score is not described in the Methods section, and it is unclear if 52% is high or low.

(10) We have added a description of the mutual information score to the Methods section (**line 702**), and note in the main text that the mutual information with unrelated, randomly generated clusters is 0%.

- L.169-172. Negative correlations over longer periods were not assessed.

(11) We had the same thought but could not think of a biological meaning of longer period negative correlations and while it gives some results, we would rather not include it since it might cause confusion.

- L.201 "frequent speciation": this is confusing since sequences were clustered at 95% nucleotide identity, yet 97%-99% ID sequences tended to be in different clusters. So what does speciation mean here? Could you discuss why it would (not) make sense to define more stringent OTU cutoffs?

(12) We hope that our modified description of the clustering explained under **point #2** above has helped clarify this point.

- L.233: mention community D.

(13) Done.

- L.236. How can I see "the passage of a hurricane system and followed by gradual, seasonal cooling of the water" in Fig. 3C and Suppl Fig 3C?

(14) We agree that this specific trend might be hard to see in Fig. 3. We now refer also to **Supplementary Figure 10**, where temperature is plotted across the entire time series. There it is more evident that the disturbance due to the passage of Earl is followed by a gradual temperature decline is clearer seen.

- L.306 Could the geographic coordinates be given in global format (rather than Google maps).

(15) Geographic coordinates are now provided (**line 386**) in degree/minute/second format, as Lat: 42° 25' 10.6" N, Lon: 70° 54' 24.2" W.

- L.316-317 states "No significant differences in community structure were detectable between the two sampling methods". This is not shown, and

- The results of replicates should be shown. Although experimental variation is briefly discussed, there is no supporting information for this and no error bars are shown anywhere (e.g. l.316 states: "No significant differences in community structure were detectable between the two sampling methods" without reference to evidence - here it is unclear what "two sampling methods" you are talking about).

(16) We thank the referee for pointing out this inaccuracy. This point has also been raised by **reviewer #1** and replied to in **point #1**.

- L.318: "Accompanying qualitative metadata include field notes, microscopy, photographs, and videos." It is unclear how this is used: if it is used then explain how, and please provide the relevant files for reproducibility. If it is not used, then this statement can be removed.

(17) We removed the statement.

- L.319. *Please include the raw data files of all the environmental parameters in a supplement.*

(18) We now provide environmental metadata values in **Supplementary File 7** and indicate this at **line 400**.

- L.322: *"each day as triplicate" - in line 314 it says only "from Day 259 forward replicates for each sample type were collected"?*

(19) Triplicates were analyzed for each day. We hope that the improved description in response to **reviewer #1, point #1** has clarified this issue.

- L.323: *Can you give a rough estimate of the time until return to the lab?*

(19) We update the indicated section (**line 409**) to provide the time between sample collection in the field and freezer-storage of processed filters, which was generally approximately 3.5 hours.

- L.327: *"collected daily and processed and analyzed" - now it looks like "daily" is only about "collected", which is to be expected. Please indicate if they were also "processed and analyzed" daily, at the end of the experiment, or in batches at some different frequency.*

(20) We modify the indicated section (**line 406**) to clarify that samples were processed up through freezing daily, and then analyzed following completion of the field sampling.

- L.382. *The potential effects of 3 rounds of PCR cycles should be discussed: the first round of PCR had an unknown number of cycles, the second had nine cycles, and the third had 45 cycles). I'm thinking of primer biases, skewing abundances, generation of chimeric sequences.*

(21) We apologize for the confusion. Indeed, we share the reviewer's concerns about cycle numbers and bias and have actually tried to minimize both. In fact, the 45 cycle PCR was only used to quantify the rRNA gene copy numbers in each sample by establishing the Ct value (i.e., the threshold cycle at which detectable amount of amplicon was present during the early exponential phase of the PCR). This allowed us to normalize the concentration of template in each sample by diluting it so that the same number of cycles could be performed while minimizing the number of cycles for amplification. Sequencing libraries were then constructed with a two-step PCR procedure: a first PCR to amplify the ribosomal gene (15 cycles for 16S amplification and 20 cycles for 18S amplification), and a second PCR with 9 cycles to add the barcodes by overlapping adapters. We also performed a QPCR to quantify amplicon concentrations for library multiplexing. We have updated the section "Ribosomal RNA gene amplification and sequencing" and specifically spell out the cycle numbers used in **line 505-510**.

- L.419. Please provide a table with Ct used.

(22) Because Ct values were used to normalize template concentration across samples, we believe that the benefit of adding a table is minimal since it would not contribute to reproducing our results.

- L 450. Statistics of the sequencing runs and sequencing QC should be provided.

(23) Quality filtering is detailed in material and methods (lines 563--568), providing the minimum quality used. We think that including the statistics in the paper might be a bit overkill but we have made a graph for the reviewer's benefit (see below).

- L.568: *"roughly centers the dynamic range of weighting around the frequencies we were able to capture given our sampling period and duration." - unclear what this means and what it is based on?*

(24) We agree that this was unclear, and apologize for the confusion. This is now clarified in the Methods section, and we copy the revised text below for reference (**line 666**).

"The parameter k can be adjusted to alter the sensitivity of the overall score to this weighting, and was chosen as 0.25 for these studies. This choice corresponds to a weight of $\frac{1}{2}$ when the difference between periods is 4 days. Since the different levels of our wavelet decomposition approximately correspond to periods of 2, 4, 8, and 16 days, this parameter value gives relatively low weights to scores between the two highest frequencies (corresponding to periods of 2 and 4 days), and relatively high weights to scores between the lowest and highest frequencies. In studies with a different sampling period and duration, the parameter can be adjusted to achieve a similar effect."

- L.596: *"we found over 99% of OTUs clustered with inflation parameter up to 2.4, with a drop in OTU inclusion above this level" - please show this.*

(25) We apologize for the lack of detail, and have added this information to the Methods section (**line 700**).

- L.623. *Could you add a brief nontrivial insight about "the method of Tado and Yamamoto"?*

(26) We have included the following clarification in the relevant portion of the Methods (**line 745**): "While the environmental time series were largely stationary, the average abundance of each cluster typically was not, and so we applied the method of Tado and Yamamoto to calculate causality. Briefly, if either time series is non-stationary, then the causality statistic does not follow its usual (chi-squared) distribution. The method corrects for this by including additional lags, up to the order of integration of the non-stationary series, in the auto-regression model."

- L.624. *For the Granger causality of the environmental drivers of a cluster, associations were considered significant at a p -value < 0.05 . However, as far as I can tell this was not corrected for multiple testing while it is unclear how many comparisons were made.*

(27) This is an excellent point that we overlooked in our initial analysis. In our revision, we correct for multiple hypotheses by considering all tests of granger causality, and calculating a false discovery rate. We now keep only those tests that pass an FDR of 10%. This is clarified in the **Methods (line 751)**, and **Figure 3** has been revised to include only the significant results.

- L.786: *"Edge width scales with p -values" - all the edges have the same width?*

(28) In revised **Figure 3** and **Supplementary Figure 3** according to previous comment we have set the same width for all edges after correcting for multiple comparisons, as just significant Granger causalities are shown.

- Figure 4 is confusing - it is unclear how to see the differences discussed in the text. Maybe rescale everything to 100% or show observed/expected ratios?

(29) We prefer to keep this format of the figure to show the distribution of interactions as genetic distance varies. However, we realize that our description led to some confusion and we have introduced some changes to clarify this figure in response to **reviewer #2 points #2 and 12**.

Also I'm very interested to see if the suggested permutation analysis (see my comment above) will show that the observation is significant: that at 97%-99% nucleotide identity closely related sequences are more frequently in different clusters than expected. I think that this is a different question than the one the authors answer with their "random community membership assignment" analysis (l.797) - but please explain this in detail in the Methods (I could not find it).

(30) This is an interesting point for testing this observation; however, the limited clustering of the permuted data does not allow for such comparison. We have extended the details in the methods section (**line 766-772**).

Reviewer #3:

Martin-Planetero et al Review.

Main comments:

Intensive time-series datasets and analysis of the combined effects of broad environmental changes with biotic interactions in the microbiome are key areas of research.

Martin-Platero and colleagues combine a novel wavelet-based community analysis method with intensive multi-marker coastal plankton time series. The results are consistent with past work, but also reveal surprisingly sharp transitions between community types that continue unabated throughout the sampling period. Thus, if you don't like the microbes present on one day, wait for the next! This observation, like Toynbee's criticism of history teaching as 'one damned fact after another', could easily devolve into meaninglessness, and I was initially afraid the manuscript would simply compile algorithmic results and high rates of community change and say 'good enough'.

Instead, the authors present a thoughtful and satisfactory narrative that both addresses broad questions about niche partitioning (lines 190-193), the evolutionary scales at which that niche partitioning occurs (Fig. 4), and the role of infrequent or 'special' events in shaping the planktonic community (see especially lines 222-257).

The novel methods presented (with a little additional testing, see below), and in particular the innovative way in which they are used in a real analysis seem both novel and broadly applicable to many systems. As much as any specific finding about this system, I find the authors handling of the tension between highly specific, partially stochastic events; long-term seasonal trends; and short-term high-frequency interactions to be exemplary, and of broad interest beyond the marine biology community. For example, investigators in the human microbiome must contend both with long-term processes like aging, but also inconsistencies in diet, infrequent antibiotic use, pathogen exposure, etc, etc. across individuals. Therefore, I think this integrative approach – in terms of both new methods and how they are applied and interpreted - will be of interest to a wide range of readers, and suitable in scope for the broad readership of Nature Communications.

I also note that the authors are careful in their handling of one potential criticism: that the changes observed by sampling one location repeatedly over time reflect both movement of water masses and microbial succession within those water masses. I do not view this potential criticism as a significant limitation of the paper as written because: a) it is similar and comparable to other oceanographic sampling b) for water-quality monitoring passage of different microbe-bearing waters by a particular site has practical consequences and c) the time-series captures both cases where this water-body-movement is likely important, and periods when broader seasonal trends dominate.

Major Comment: More quantification needed on the interaction of distribution-based

clustering and wavelet analysis

There is currently a split in the field between sub-OTU methods that operate on a per-sample basis, and those that use information on distribution across samples. For some study questions, this split probably doesn't matter too greatly, as broad trends in taxonomic change or beta-diversity will likely emerge regardless. For the more nuanced questions of temporal association tested here, I found myself wondering whether a flexible clustering criterion defined based on distribution is problematic.

If we define OTUs based on ecological cohesiveness, and then test a hypothesis about ecological cohesiveness over time, are we predisposing the results to favor sharp turnovers and/or correlated outcomes to a greater degree than if we used a per-sample method (perhaps deblur)?

(1) This is an interesting point – thank you for raising it. We think deblur is a great method and we have recently started to use it. However, the two methods are similar in spirit in that they try to minimize contributions by sequencing errors that are clustered around 'true' sequences. We respond to the Reviewer's specific suggestions below.

*In other words, is it possible to show (or have you already shown) that the false-positive rate for detection of cohesive communities is not increased when using distribution-based clustering? If so, can we 'subtract out' these effects from the observed trends?
Page 4 (lines 86-89).*

Here's what I'm thinking: take the 'conventional' OTUs already generated for relatedness analysis (e.g. lines 632-636) or a non-distribution based sub-OTU method like deblur (in a perfect world, perhaps both). Compare these to the distribution-based OTUs. Use a permutational procedure to destroy real temporal signal and co-association in each collection of OTU tables. Then apply wavelet analysis and look for false-positive associations. In particular, I am interested in a quantification of whether there are more FPs in the distribution-based OTUs than the conventional OTUs or deblurred sequences, and whether the OTU clustering threshold has a strong effect?

I appreciated the existing simulations testing detection power for the wavelet-based approach used here vs. e.g. Pearson correlation. Regardless of the outcome, a better quantification of these interactions would strengthen the paper.

(2) To address the Reviewer's concern (which is related to a major comment from **Reviewer #2, point #2**), we performed **new analysis** to assess the statistical significance of the wavelet similarity scores. We asked if there are more similarity scores falling below significance (false positives) in data clustered (non-distributionally) at 100% OTU identity, compared with the original data. We found that the rate of false positives was nearly identical using either method, and that clustering at 100% OTU identity resulted in a similar number of clusters with sharp turnovers and similar abundance patterns (**Supplementary Figure 6**). We describe this analysis in our revised **Results (line 193-227)** and **Methods**, and also provide details below.

To assess the frequency of statistically significant similarity scores in both distributional and non-distributional clustered OTUs, we performed a permutation analysis. Specifically, we permuted each time series independently and calculated wavelet similarity scores, noting for each pair of OTUs if the scores were greater than those observed in the unpermuted data. We repeated these permutations 50,000 times, and used the results to calculate empirical p-values.

We found that the original data have a clear enrichment for significant p-values using both distributional OTUs (shown in **new Supplementary Figure 6A**) and non-distributional OTUs (**Supplementary Figure 6B**). We then used these p-values to identify similarity scores that pass a false discovery rate (FDR) of 10%. Clustering the distributional OTUs using only those similarity scores passing the FDR test results in clusters that are in good agreement with our original results (55% mutual information), and clusters of non-distributional OTUs have very similar patterns, with a similar number of clusters, sharp transitions, and similar patterns of abundance (shown in **new Supplementary Figure 8**). We also note that using a conservative FDR test, only 5% of the scores that passed the original thresholds also pass the FDR test (with both distributional and non-distributional OTUs). Thus, overall cluster structure remains robust, even if less stringent connections between nodes are allowed.

Minor comments:

Line 24. How do we feel about using 'protozoa'? Would microbial eukaryotes (e.g. including oceanic fungi) or even protists (same idea but without the connotation of primitive animals) be more appropriate?

(3) Agreed. Changed to protists.

Line 28. Do we regard plankton a countable noun? If so, consider Is are.

(4) Agreed. Changed.

Line 36. 'in spite of physical instability in the coastal ocean, defined communities of organisms assemble'. This conclusion to the abstract seemed out of place to me, and a bit confusing in terms of it's actual meaning. On the one hand, 'defined communities of organisms assemble' begs the question of what the other alternative is. I think it could be sharpened in order to not push readers out before they have a chance to get to some of the neat analysis later in the paper. Setting aside the phrasing, isn't this the opposite of your conclusion? On the other hand, Based on the network analysis and hypotheses presented later, isn't it precisely the physical instability of the ocean (warm temperature water bodies; sediment-bearing upwelling events; seasonal cooling, etc.) that allows the sharp community transitions reported here to occur?

(5) Very good point. We changed to: "Our analysis thus highlights that the physical instability of the coastal ocean is mirrored in sharp transitions of defined but ephemeral communities of organisms."

Line 99-105. Haven't these 'seed bank' dynamics also been reported in several other studies that should be cited? E.g. In the Western English Channel (Caporaso et al., 2012; <https://www.ncbi.nlm.nih.gov/pmc/articles/PMC3358019/>), and indirectly by comparison of deep sequencing of single samples to global samples from ICoMM (Gibbons 2013; <http://www.pnas.org/content/110/12/4651.short>). I should emphasize that I do not mention these other studies because I think they cut into the novelty of this study (which I think lies elsewhere) but merely because I think they are an important context that might help reinforce or refine the interpretations here.

(6) Thank you for this point. We have added the appropriate citations.

Lines 190 – 193. I thought this was a really nice passage relating the specific observations described here (rapid turnover of associated communities) to broader ecological questions about diversity and niche space. I wonder if there is a way to work a short version of this idea into the abstract (perhaps in place of the last sentence?)

(7) Thank you for this point. We tried to incorporate these thoughts into the abstract but found them ultimately too complicated to fully explain.

Lines 222 – 257 I found this final passage quite remarkable. It highlights quite effectively the intersection of oceanographic, biological, and meteorological processes that we must deal with in trying to understand real natural systems. Often, some of the factors that matter most – are messy, surprising, and perhaps not innately appealing to the simple stories we as humans like to tell (and the simple hypotheses we like to test). The combination of a newer method for community analysis, along with this thoughtful narrative of broader processes affecting the environment seems closer to a real understanding of the system than tests of any one single-factor hypothesis considered in isolation. These observations also speak to the value of intensive time-series datasets, and suggest many jumping-off points for subsequent studies.

(8) We thank the reviewer for the nice words.

REVIEWERS' COMMENTS:

Reviewer #1 - Primary reviewer unavailable, but co-reviewer provided a continuation of the original comments as Reviewer #4.

Reviewer #2 (Remarks to the Author):

The authors have made a good effort to improve the manuscript: most points have been addressed or sufficiently rebutted. Perhaps they could still discuss/define "speciation" a bit more explicitly in line 264: how are different "species" defined? As diverging strains having non-overlapping functional repertoires, or something else? Is it important here how we understand bacterial species demarcation? It might be good to state explicitly what you call "speciation" here. I think the paper will prove to be exciting reading for many.

Reviewer #3 (Remarks to the Author):

Thank you for your thoughtful responses. The additional analyses and revisions conducted address my concerns.

Reviewer #4 (Remarks to the Author):

Review of revised manuscript by Martin-Platero et al.

Overall comments:

I would like to thank the authors for making earnest efforts to address the critiques presented in the first round of reviews. I think the manuscript has been improved substantially, especially in its handling of interacting biotic and environmental drivers. I agree that studying interactions between microbes is important, and should not be put off simply because methods to untangle environmental versus biotic drivers are imperfect. I also appreciated the new sensitivity analysis of WaveClust, which indicates to future users the conditions where the method should perform best. It is encouraging to see that the analyses conducted on the first differenced data are similar to the original analyses. The following critiques are aimed at clarifying the strength of the evidence supporting various interpretations of the data.

Major comments:

Lines 181-191: I realize that a similar comment I made in the first round of review was not specific enough to be helpful, so I will try to reword. I agree that this interpretation is a plausible explanation of the observed results. However, without a model or metadata, the certainty ascribed to this hypothesis is currently too high. This is especially true because of the placement of this paragraph in the results section, which is often considered to be less speculative than the discussion section. Please rephrase to indicate that other explanations may exist. For instance, lines 181-184:
"One explanation is that there were characteristic high frequency negative correlations (possibly indicating predation or competition) between pairs of OTUs occurring over long time periods, but that high frequency positive correlations (possibly arising from cooperation) were temporally more limited"

The above text is not intended to be inserted verbatim, but instead is given as an example of the statement of the degree of certainty about the proposed explanation. The authors noted in their rebuttal that it is common for interaction and correlation to be used interchangeably; while that is

true, I think it is important to spend the extra time to be more accurate, especially in studies that are expected to be widely read.

Minor comments:

One strong piece of evidence that the clusters are cohesive communities would be observing re-emergence of similar clusters in subsequent years. Perhaps this point could be made more explicitly in the paragraph on lines 346-356 in the discussion.

On lines 623 – 625: Currently, this statement reads as if biotic interactions are not expected to be important on long time scales. Traditional ecologists would likely take issue with that interpretation. It might be more inclusive to rephrase to indicate that interactions are expected to be important on multiple time scales, including short ones

Response to reviewers' comments.

We again thank all the reviewers for the comments, which we feel have greatly improved the manuscript.

Reviewer #1:

Reviewer #2:

The authors have made a good effort to improve the manuscript: most points have been addressed or sufficiently rebutted. Perhaps they could still discuss/define "speciation" a bit more explicitly in line 264: how are different "species" defined? As diverging strains having non-overlapping functional repertoires, or something else? Is it important here how we understand bacterial species demarcation? It might be good to state explicitly what you call "speciation" here. I think the paper will prove to be exciting reading for many.

Regarding species and speciation, we note that speciation is a process that does not necessarily require new species to arise. Most relevant here is that ecological tradeoffs may lead to a separation in gene pools. We have attempted to better explain what we mean:

“In fact, such differential associations may be a direct outcome of the speciation process since the formation of distinct lineages in a highly interconnected environment like the Ocean requires the evolution of ecological tradeoffs, which induce spatial and/or temporal separation of gene pools and hence allow differential ecological specialization to evolve and adaptive traits to spread in a population-specific manner 31.”

Reviewer #3:

Reviewer #4:

Overall comments:

I would like to thank the authors for making earnest efforts to address the critiques presented in the first round of reviews. I think the manuscript has been improved substantially, especially in its handling of interacting biotic and environmental drivers. I agree that studying interactions between microbes is important, and should not be put off simply because methods to untangle environmental versus biotic drivers are imperfect. I also appreciated the new sensitivity analysis of WaveClust, which indicates to future users the conditions where the method should perform best. It is encouraging to see that the analyses conducted on the first differenced data are similar to the original analyses. The following critiques are aimed at clarifying the strength of the evidence supporting various interpretations of the data.

Major comments:

Lines 181-191: I realize that a similar comment I made in the first round of review was not specific enough to be helpful, so I will try to reword. I agree that this interpretation is a plausible explanation of the observed results. However, without a model or metadata, the certainty ascribed to this hypothesis is currently too high. This is especially true because of the placement of this paragraph in the results section, which is often considered to be less

speculative than the discussion section. Please rephrase to indicate that other explanations may exist. For instance, lines 181-184:

“One explanation is that there were characteristic high frequency negative correlations (possibly indicating predation or competition) between pairs of OTUs occurring over long time periods, but that high frequency positive correlations (possibly arising from cooperation) were temporally more limited”

The above text is not intended to be inserted verbatim, but instead is given as an example of the statement of the degree of certainty about the proposed explanation. The authors noted in their rebuttal that it is common for interaction and correlation to be used interchangeably; while that is true, I think it is important to spend the extra time to be more accurate, especially in studies that are expected to be widely read.

We agree and have followed the suggestion very closely.

“This indicates that there were characteristic high frequency negative interactions (possibly due to competition or predation) between pairs of OTUs occurring over longer time periods but that high frequency positive interactions (possibly due to cooperation) were temporally more limited. One explanation of the latter would be that shorter lived blooms of organisms exploiting similar resources may lead to metabolic interactions (Fig. 3A and Supplementary Fig. 3A).”

Minor comments:

One strong piece of evidence that the clusters are cohesive communities would be observing re-emergence of similar clusters in subsequent years. Perhaps this point could be made more explicitly in the paragraph on lines 346-356 in the discussion.

We note that one paragraph in the discussion already had a question about community reassembly. We have modified the statement to more directly reflect the reviewer’s concern.

“These observations pose the intriguing question to what extent communities reassemble across longer time periods as further evidence for their cohesiveness and whether they assemble reproducibly enough for their OTU composition to serve as an accurate indicator of environmental conditions.”

On lines 623 – 625: Currently, this statement reads as if biotic interactions are not expected to be important on long time scales. Traditional ecologists would likely take issue with that interpretation. It might be more inclusive to rephrase to indicate that interactions are expected to be important on multiple time scales, including short ones

We agree. However, in the microbial world much of the chemical regime is caused by other organisms (e.g., blooms of primary producers). We attempted to clarify.

For example, overall physical and chemical regimes are expected to prevail over longer periods suggesting that pairs of organisms responding by growth to these conditions are positively correlated over longer periods (lower frequencies). Moreover, such growth (e.g., by characteristic primary producers) may lead to longer-lasting interactions.